# Enhancing Interpretability: A Versatile Clue-Based Framework for Faithful In-Depth Interpretations

## Abstract

Despite the state-of-the-art performance of deep neural networks, they are susceptible to bias and malfunction in unforeseen situations. Moreover, the complex computations underlying their reasoning are not human-understandable, hindering the development of trust and the validation of decisions. Local explainable AI (XAI) methods seek to provide explanations for individual model decisions with two key goals: faithfulness to the model and human-understandability. However, existing approaches often suffer from performance loss, limited applicability to pre-trained models, and unfaithful explanations. Seeking interpretations with higher fidelity, we introduce a novel definition, called Distinguishing Clue, which is a set of input regions that uniquely promote specific network decisions, detected through our Local Attention Perception (LAP) module. Our innovative training scheme allows LAP to learn these clues without relying on expert annotations. It also provides a means for injecting both general and expert knowledge. The system is usable for training networks from scratch, enhancing their interpretability, and explaining networks that have already been trained. We demonstrate the superiority of the proposed method by evaluating it on different architectures across two datasets, including ImageNet. Our results show that the proposed framework offers interpretations that are more valid and more faithful to the model than those produced by baseline XAI methods.

## 1 Introduction

Artificial Intelligence (AI) has found real-world applications in clinical computer-aided decision systems, medical diagnosis, and autonomous driving. These critical applications raise concerns about whether AI models are trustworthy and whether their decisions are valid (Tjoa & Guan, 2020). Deep Neural Networks (DNNs), among the most successful AI models, make decisions using complex computations that humans do not understand. They are trained end-to-end and are susceptible to learning detours and biases from the dataset rather than the actual concepts. Since AI is now responsible for decisions affecting human rights and ethics, governments have begun enacting laws requiring explanations in certain situations (Goodman & Flaxman, 2017). This has increased the demand for DNNs to explain themselves. Explaining DNNs offers benefits beyond decision verification, including bias detection, trust development, and legal compliance (Caruana et al., 2015); it also helps diagnose and improve the model (Weber et al., 2023; Anders et al., 2022; Gupta et al., 2023). Additionally, knowledge can be discovered from models with superior-than-human performance to enrich human understanding (Du et al., 2019).

Interpreting [1] the decisions of Deep Neural Networks (DNNs) at the level of individual samples has gained attention in recent years, with local XAI methods providing insights into single predictions. There are two paradigms for local XAI: intrinsic methods and post-hoc methods. Intrinsic interpretability is achieved by enforcing interpretability directly into the model's architecture and training strategies (Zhang et al., 2018; Chen et al., 2016; Freitas, 2014), enabling the model itself to provide explanations for its decisions. However, these methods do not apply to already trained models (Gilpin et al., 2018; Kim et al., 2018) and often

---

[1] Throughout this paper, we use the verb 'interpreting' in a neutral sense to refer to both intrinsic interpretability methods and post-hoc explainability methods, unless otherwise specified.

impose architectural constraints or trade-offs with performance (Gilpin et al., 2018). Post-hoc methods, in contrast, provide explanations for models that have already been trained by adopting assumptions and simplifications, which can sometimes lead to misleading explanations, as they may not be valid in the model's decision-making process (Gilpin et al., 2018; Sixt et al., 2020). Most local XAI methods provide interpretations based on the last few layers of the network, which, although insightful, are inherently low-resolution and fail to precisely localize the input regions responsible for the model's decisions; furthermore, they are insufficient for supporting comprehensive diagnostic analysis of the model. While some methods attempt to provide interpretations based on the middle layers, their interpretive performance typically degrades as interpretations move farther from the decision layer. Independent of the paradigm, there are two crucial objectives in local XAI: **faithfulness** and **human-understandability**. Faithfulness means that explanations accurately reflect the valid reasons behind the model's decisions, capturing the underlying decision-making process. Human-understandability, on the other hand, requires explanations to be easy to comprehend. Given the need for human-understandability, recent advancements in XAI have focused on providing explanations in terms of human-understandable concepts, defined as semantically meaningful groups of input features that are effective in the model's decision-making, rather than single attribution maps of raw features (Schwalbe, 2022). While the initial methods depended on expert-defined concept annotations (Kim et al., 2018; Graziani et al., 2018; Koh et al., 2020), unsupervised local concept-based methods have developed approaches for providing explanations without requiring expert annotations (Ghorbani et al., 2019; Zhang et al., 2021; Fel et al., 2023; Wang et al., 2024). These approaches involve identifying regions that influence decision-making, defining concepts based on these regions across the entire dataset, and expressing these concepts in a human-comprehensible manner, for example, using word phrases (Radford et al., 2021; Ahn et al., 2024; Oikarinen & Weng, 2022). Potential regions are detected using various techniques such as super-pixel segmentation (Ghorbani et al., 2019), random cropping (Fel et al., 2023), object segmentation models trained on large datasets (Kirillov et al., 2023; Sun et al., 2023), and other local explanation methods (Wang et al., 2022). While the primary focus of these studies has been concept definition for human-understandability, accurately localizing the input regions influencing the model's decision-making is also a critical step. This step leads to extracting more meaningful concepts and providing explanations that are reliable and sound, aligning with faithfulness as the other core objective of local XAI methods.

This study directly addresses the practical need for a versatile method to detect the input regions, that are crucial to the model's decision-making process. To this end, we developed a technique with a novel perspective by introducing Distinguishing Clues (DCs) as a set of input regions that favor a subset of network outputs over others. Our framework for local interpretation introduces a set of DCs that distinguish each output from others, along with a module called Local Attention Perception (LAP) that identifies the input regions associated with each DC at any desired network layer, providing in-depth interpretations. Additionally, we propose a new training scheme that enables the training of LAPs without requiring detailed annotations, making the approach broadly applicable to any dataset without additional requirements. Furthermore, this training scheme allows existing knowledge about the task, e.g., the minimum size of distinguishing regions, to be injected into the model's training, thereby enhancing its interpretability. Accordingly, LAP is a versatile XAI method: it functions as an intrinsic interpretability approach during training (through auxiliary loss terms that shape the network's internal representations and by directly contributing to the decision-making process when used as a plugged-in module), and it can also be applied as a post-hoc explainer for already trained networks without any retraining. We assessed the superiority of the proposed framework to other local XAI methods under two settings: training a model from scratch and explaining an already trained model. The results demonstrate that LAPs can provide a more faithful-to-the-model interpretation than other local XAI methods, without causing a performance drop in either setting. We also demonstrated the effectiveness of the provided training scheme through extensive ablation studies. The key contributions are:

- Introducing a versatile framework for local interpretation of DNNs that enhances interpretability without constraining architecture or compromising performance, applicable to both training from scratch and trained models, surpassing renowned local XAI methods.

- Defining a *System of Distinguishing Clues* that provides local interpretations, promoting each output at different depths of the network, allowing depth-wise diagnosis of the model.

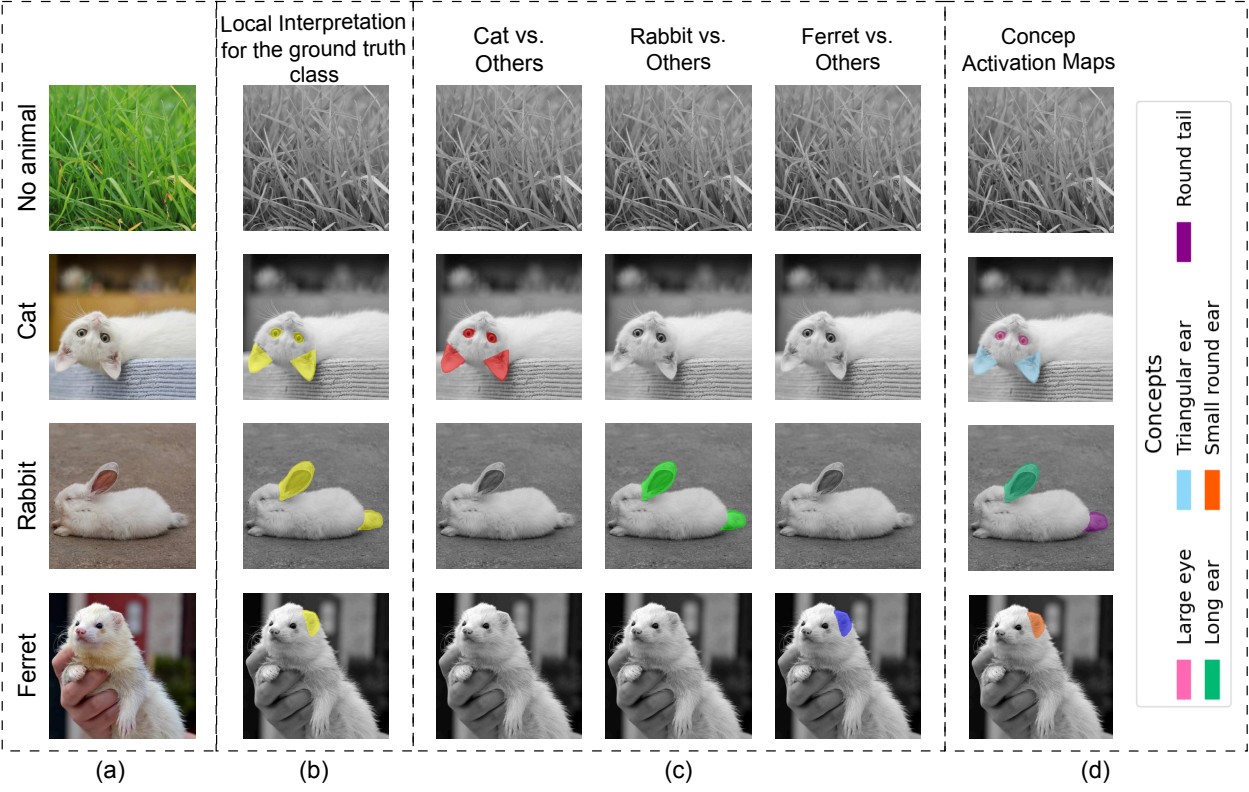

Figure 1: **(a)** Representative images from a 4-class classification task: No Animal (Pixnio, n.d.), Cat (FreeRangeStock, n.d.), Rabbit (Timeless Moon, n.d.), and Ferret (Pexels, n.d.). **(b)** For a network trained on this task, a local XAI method scores input pixels based on their contribution to the network's decision for a specific class, providing attribution maps. Notably, local interpretation differs from segmentation; shared features, such as a furry body, receive low scores in any single class. **(c)** An example clue system for interpreting the network considers three Distinguishing Clues (DC), Cat vs. Others, Rabbit vs. Others, and Ferret vs. Others, each scoring pixels contributing to their respective classes. The "No Animal" class serves as the Background output (BGO), defined by the absence of clues for other classes. The training scheme proposed in this paper enables learning these DCs without expert annotations. **(d)** Local concept-based XAI scores pixels favoring each class and represents them in a human-understandable form, such as naming. Using more precise local attribution maps as input to concept-based methods can lead to the discovery of more effective and meaningful concepts for interpreting network decisions. b, c, and d show sample ideal attribution maps for a hypothetical trained network.

- Proposing a novel training scheme for detecting the DCs without depending on experts' annotations that provides a means for general knowledge injection in training.

- Designing a module for the localization of DCs in the DNNs, usable as both a detached module and a plugged-in module inside the architecture, even in the already trained models, without negatively affecting the performance of the main task.

## 2 Method: LAP-Based Interpretation

In this section, we introduce LAP-based interpretation as our method. For this purpose, we first define Distinguishing Clue (DC).

**Definition 1** *A Distinguishing Clue (DC) is a set of input regions whose presence has a positive effect on the model's decision toward a subset of outputs (classes) relative to others. Here, an input region refers to a contiguous segment of the input (e.g., a group of pixels in an image).*

Furthermore, when no distinguishable input regions can be identified that explicitly contribute to a given output class, we define the class as **BGO (Background Only)**, indicating that no corresponding DC is required for its interpretation. Since the BGO class represents the absence of class-specific evidence and captures only background responses, there can be at most one such class.

Note that DNNs implicitly learn task-specific discriminative features, progressively refining them with increasing depth through the layers. While the final layers capture the most definitive representations, intermediate layers (or section[2]) develop earlier, less precise understandings. Therefore, we leverage DCs (Definition 1) to identify input regions associated with these features at any network section. These regions serve as local interpretations for the learned representations in DNN networks up to that section. We propose a module called LAP that computes importance probabilities for each element per DC. Elements refer to components of the input data or intermediate feature maps, where each component is represented by a feature vector, such as pixels with RGB channels, voxels in 3D images, or feature map locations, each corresponding to a patch in the input image. A flexible training scheme trains LAPs either as post-hoc explainers for trained networks or as plug-in modules that provide self-interpretability, enabling LAPs to score DC-wise importance probabilities for each element. While applicable to inputs of various dimensions and modalities, we demonstrate its use for 2D inputs (pixels) in this paper, with the formulation seamlessly extending to any dimensionality or modality. Next, we provide an example for DC and compare it with a similar concept-based XAI method.

**Example 2.1** *Samples of a four-class classification task, with classes "No Animal," "Cat," "Rabbit," and "Ferret," are presented in Fig. 1. a. For these samples, local interpretations (Fig. 1.b), DC-based interpretations (Fig. 1.c), and concept-based interpretations (Fig. 1.d) are provided for a hypothetical model trained on this task. For this example, we have employed a set of three DCs to provide interpretations: "Cat vs. Others," "Rabbit vs. Others," and "Ferret vs. Others" (Fig. 1.c). Each DC assigns scores to input regions that promote its associated class over the others. For example, input regions containing features that promote the class "Cat" over other classes, e.g., triangular ears and large eyes, constitute the DC for "Cat vs. Others." For the "No Animal" class, no distinguishing input regions can be identified that explicitly promote it; instead, the absence of discriminative evidence for the other classes leads the network to favor it. This is called a BGO (Background Only) class in DC-based interpretation and therefore does not require a corresponding DC for interpretation. The DCs together provide the local interpretation for each desired target label.*

*The DC-based interpretation is also compared with concept-based XAI methods, which score pixels favoring each class and represent them in a human-understandable form, such as naming (Fig. 1.d). Both approaches belong to the domain of local XAI methods and therefore share the common objective of scoring input pixels that promote each target label, making their methods comparable in terms of the validity of the identified pixels. Concept-based methods add an additional step that presents highly scored regions in a human-understandable way.*

The following sections detail the definition of DCs, the LAP architecture, and the training scheme; additionally, a summary of key terms is provided in Appendix (App) 5.

## 2.1 A Complete System of DCs for Discriminating Between outputs

To formalize the problem setting, consider an input space $\mathbb{R}^{F \times H \times W}$ where $F$, $H$ and $W$ denote input per element's features, height and width, respectively. We also denote $N$ possible outputs as $Y^{1:N}$, and a dataset of input-output pairs $D = \{(X_i, y_i) \in \mathbb{R}^{F \times H \times W} \times Y^{1:N}\}$. Adopting an element-based perspective, we represent each input as a set of elements, where each element is a pair consisting of a spatial location and its corresponding feature vector (i.e., ($\mathbb{R}^F$)). As mentioned in Definition 1, a **DC** is a combination of elements

---

[2]We define a section as a width cut that may contain multiple layers belonging to parallel paths in the model architecture, jointly capturing the representational capacity of parallel components at a given network depth.

corresponding to discriminative features that promote specific network outputs over others. Formally, each DC, $\text{dc}_m$, promotes a subset of outputs ($Y^{\text{dc}_m^+} \subset Y^{1:N}$) over another exclusive subset ($Y^{\text{dc}_m^-} \subset Y^{1:N} \setminus Y^{\text{dc}_m^+}$). According to Eq. (1), each $\text{dc}_m$ is the set of combinations of elements present in inputs related to $Y^{\text{dc}_m^+}$ but absent from those related to $Y^{\text{dc}_m^-}$. Here, $\mathbb{P}_{H \times W}(\mathbb{R}^F)$ denotes the power set of possible input elements, $\wedge$ and $\vee$ denotes AND and OR functions.

$$
\begin{aligned}
\text{dc}_m = \Big\{ z \in \mathbb{P}_{H \times W}(\mathbb{R}^F) | & \big( \exists (X, y) \in (\mathbb{R}^{F \times H \times W}, Y^{1:N}) : (y \in Y^{\text{dc}_m^+}) \wedge (z \subseteq X) \big) \\
& \wedge \big( \nexists (X, y) \in (\mathbb{R}^{F \times H \times W}, Y^{1:N}) : (y \in Y^{\text{dc}_m^-}) \wedge (z \subseteq X) \big) \Big\}.
\end{aligned}
\tag{1}
$$

For a comprehensive understanding of the rationale behind an output in the network, the system of defined DCs ($\text{DC}^{1:M}$) must satisfy the completeness condition as specified in Eq. (2) which holds $\forall y \in Y^{1:N}$. To satisfy completeness, either one output ($y$) is designated as the Background output (**BGO**), or there exists a subset of DCs ($\text{DC}_y$) that uniquely distinguish $y$ from all other outputs. The **BGO** denotes an output for which no distinguishing clue can be identified; the absence of clues for alternative outputs results in the network selecting the BGO (see Example 2.1). For instance, in binary existence classification, the negative class functions as the BGO, and a single DC distinguishing the positive class from the BGO suffices. Likewise, for an $N$-class classification task, a one-vs-others scheme can establish a complete set of DCs (see Example 2.1).

$$
\Big( (y = \text{BGO}) \wedge (\nexists y' \in Y^{1:N} : (y' \neq y) \wedge (y' = \text{BGO})) \Big) \vee \Big( \exists \text{DC}_y \subset \text{DC}^{1:M} : (( \bigcap_{\text{dc}_m \in \text{DC}_y} Y^{\text{dc}_m^+}) = \{y\}) \Big),
$$
$$
\forall y \in Y^{1:N}.
\tag{2}
$$

## 2.2  In-Depth LAP Units for DC Localization

**LAP:** We designed a module called **Local Attention Perception (LAP)** to automatically identify DCs at any network layer. LAP generates probabilistic importance maps that highlight which pixels contribute most to a decision at different network depths. LAP can be viewed as an attention mechanism, since it assigns importance scores to input pixels. However, unlike standard attention, it operates on each pixel's feature vector alone, without explicit key-value pairs, and where "importance to a DC" implicitly serves as the "key" [3]. As shown in Fig. 2.a, an LAP is a trainable unit that takes the feature map from the section of the network we aim to interpret ($X^l$) and computes DC-specific importance probability maps for that section ($I^l$), mathematically expressed in Eq. (3) for the $l$-th LAP. For a system with $M$ DCs, the LAP has $M$ output heads, each estimating the relevance probability of a pixel to one DC ($I_{ij}$). In this design, probabilities are used instead of raw importance scores because they are more intuitive for humans and enable the use of loss functions for training LAPs, as further detailed in Sec. 2.3. One possible LAP implementation uses stacked convolutional layers with $1 \times 1$ kernels and $M$ output channels, followed by a sigmoid activation to generate the final importance probabilities.

$$
I_{ij}^l = \text{LAP}^l(X_{ij}^l); \qquad \text{LAP}^l : \mathbb{R}^F \to [0, 1]^M
\tag{3}
$$

**LAP-Pool:** Since LAP generates DC-wise importance maps that highlight key task-specific input regions, these maps can be leveraged within pooling layers as a practical extension of our framework, though this does not constitute the main focus of this work. As illustrated in Fig. 2.b, LAP-Pool aggregates the importance maps computed by LAP into a unified pixel-wise importance score map. For each kernel position in the pooling process, the scores for pixels within the kernel are normalized, and the final feature vector is obtained by weighted averaging these normalized scores. This approach mimics a dynamic zooming mechanism that identifies the most critical pixels under each kernel and propagates their features rather than indiscriminately mixing them, as in average pooling. Consequently, important fine-grained details are preserved and propagated through the network's depth while avoiding artifacts from naive feature mixing. For greater clarity, we have

---

[3]The differences are discussed in detail in App. C

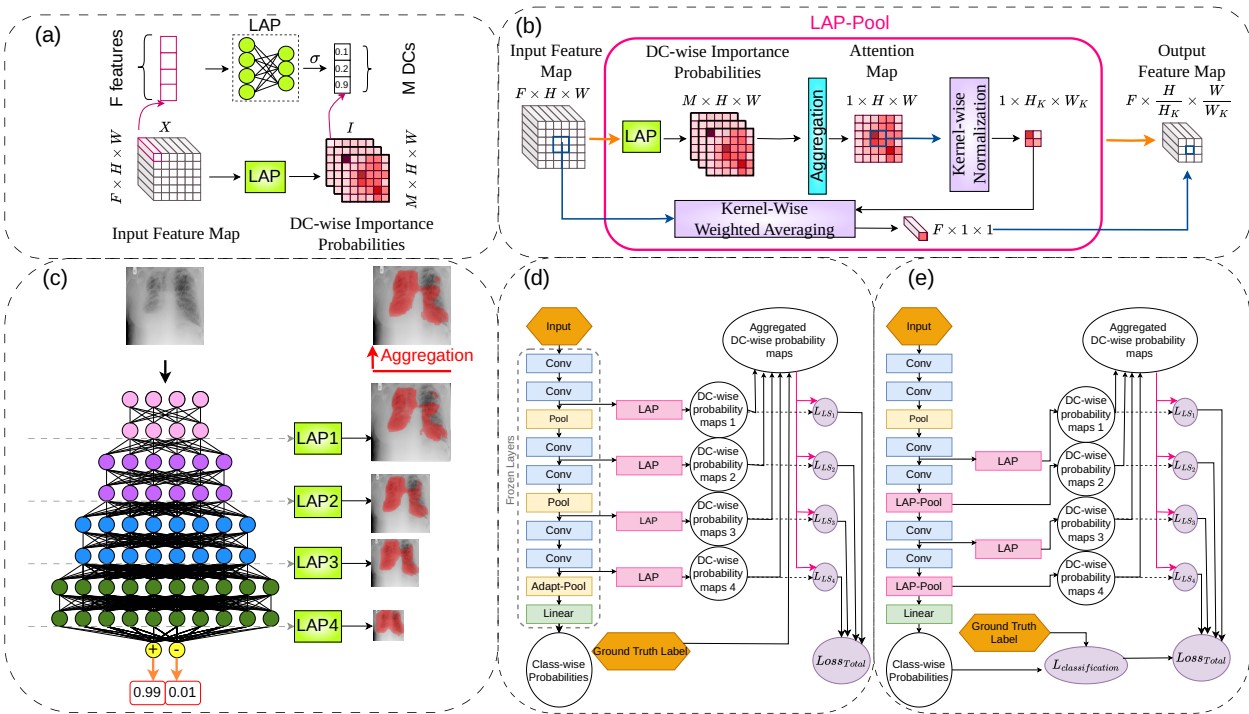

Figure 2: **(a)** Architecture of a Local Attention Perception (LAP) unit. LAP takes a feature map as input and outputs a probabilistic importance map that highlights the regions most relevant to each Distinguishing Clue (DC). **(b)** Architecture of a **LAP-Pool** module. LAP-Pool aggregates the probabilistic importance maps of all DCs into a single importance score map, which guides pooling by weighting pixel features based on their importance, preserving the features from the most informative regions. **(c)** LAP units can be applied at various depths of the network to provide sectional and complementary interpretations. LAPs closer to the input produce higher-resolution maps, while those near the output layer yield more decision-relevant interpretations. By aggregating outputs from multiple LAPs, a final interpretation can be constructed that balances spatial detail with decision-level accuracy. **(d)** and **(e)** LAPs can be used in two distinct paradigms for interpreting networks. In the post-hoc explanation setting (**d**), LAPs analyze and explain pre-trained networks without modifying their architecture or weights; here, the network parameters are frozen, and only the LAP modules are trained using LAP-specific supervision losses. Alternatively, in the intrinsic interpretation paradigm (**e**), LAPs are integrated during training, allowing both the network and LAP modules to be trained jointly from scratch. In this approach, LAP units can be implemented either as plug-in LAP-Pools or as detached modules at arbitrary points within the network. Combining LAP supervision losses with the main classification loss encourages the network to learn features that explicitly reveal distinguishing clues, resulting in a model that is inherently more interpretable.

adopted the unified pooling framework of Gao et al. (Gao et al., 2019) to formalize the LAP-Pool procedure in App. C.1.

**Aggregation through the depth:** As stated earlier, final layers capture the most definitive task-specific features, while intermediate layers develop earlier, less accurate understandings. A single LAP module provides pixel-wise importance probabilities per DC at a chosen network section, reflecting the level of understanding up to that layer. Multiple LAP units can be employed to obtain DC-wise importance maps at different network depths, yielding maps with varying resolutions and accuracies, as illustrated in Fig. 2.c. Maps closer to the final decision layer tend to be more accurate, but have lower resolution and less precise spatial alignment with the input image. Conversely, maps nearer the input layer offer higher resolution but are less accurate. To leverage the advantages of both, the DC-wise probability maps are combined across the

network depth, yielding a single, refined local interpretation. In the proposed aggregation approach, pixel-wise importance probability maps are aggregated independently for each DC. A summary of the aggregation procedure for a given DC is as follows. The presence of a clue (i.e., an importance probability greater than 0.5) is first determined from the lowest-resolution map, which is closest to the decider. Each pixel in the low-resolution map corresponds to non-overlapping square regions of pixels in the higher-resolution maps. If a clue is present in a low-resolution pixel, the corresponding pixels in high-resolution maps are examined: if any pixel within that region also contains a clue, the location of the clue is sharpened using the higher-resolution information; otherwise, the entire region inherits the probability value of the low-resolution pixel. The full algorithm is described in detail in App. C.2.

**Usage:** LAP modules can be employed in two ways: as detached LAPs, which receive the feature map and compute DC-wise importance probability maps without altering the network's forward flow, or as plugged-in LAP-Pools (Fig. 2 d and e). LAP-Pools can be integrated into any convolutional architecture by replacing pooling or adaptive pooling layers. Strided convolutions can likewise be substituted with a stride-one convolution followed by a LAP-Pool matching the original kernel size and stride. LAPs support various training and deployment scenarios: detached LAPs can be applied to explain trained models without altering their operation (Fig. 2.d). Alternatively, LAP-Pools and LAP modules can be trained jointly with networks from scratch in an end-to-end manner, as shown in Fig. 2.e. Additionally, LAPs enable a selective fine-tuning setting, where LAP-Pools are integrated into pre-trained models and optimized either independently or alongside specific network components, while freezing the remaining parameters. This approach leverages the selective nature of pooling operations as learned feature-stream selectors, facilitating the efficient adaptation of existing architectures. A general pseudo-code for training LAPs in both modes, joint training with the network and detached training while keeping the backbone network frozen, is provided in App. C.4. Additionally, we provided a detailed discussion of LAP-based interpretation from the fidelity and faithfulness perspectives for both approaches in App. C.3.

### 2.3 Weakly Supervised Training Scheme for Learning DCs

**Practical Realization of the Formal Definition:** According to Definition 1, a DC is a set of input regions that jointly favors a subset of classes (DC+) over another subset (DC-). These regions should appear in samples from DC+ classes and be absent from DC- classes. Importantly, their discriminative value may arise from their combination rather than from isolated parts. The practical objective is therefore to recover, without ground-truth localization, the regions that collectively instantiate each DC.

In practice, a neural network represents an input through a hierarchy of feature maps, where each element corresponds to a receptive-field patch in the original input. As the network becomes deeper, receptive fields expand, allowing later layers to encode increasingly complex spatial relationships. To operationalize the formal definition, we approximate the problem of identifying a complete set of regions forming a DC by identifying, across different network depths, the individual feature-map elements associated with that DC. This is the first gap between the formal and practical settings. A second gap arises because shallow layers observe only fragments of a DC, whereas deeper layers capture more complete, but still imperfect, evidence due to representation and optimization limitations. We therefore replace the strict presence/absence criterion of the formal definition with a probabilistic formulation that estimates, for each element, the likelihood of being related to a DC. Elements whose feature vectors occur frequently in DC+ samples and only rarely in DC- samples should receive higher probabilities.

Ideally, learning such probabilities would require element-wise binary labels indicating whether each feature-map element belongs to a DC. Since such annotations are unavailable, we adopt a weakly supervised training procedure in which LAP units provide imperfect DC estimates that serve as semi-ground-truth in subsequent iterations. Because feature maps remain spatially aligned across layers, LAP units at different depths can reinforce one another during this bootstrap process. At the same time, not all elements in a feature map are DC-related: many correspond to background patterns shared across classes, and the number of DC-related elements varies across samples and is unknown. To reduce supervision noise under this uncertainty, we label only the most reliable elements. Concretely, we focus supervision on a selected subset of elements: those most likely to be DC-related, those most likely to be background-related (i.e., least likely to belong to the DC), and, in DC-absent samples, the most challenging high-scoring negatives. This selective-labeling strategy

generates semi-ground-truth labels only where the model is most confident. To learn DC-related elements under this selective-labeling regime, we define three loss components:

- **Coverage:** In a DC-present sample, at least some feature-map elements should be related to the DC. The **Min Active Ratio (MinAR)** term encourages LAPs to assign high importance probabilities to such elements.

- **Precision:** Even in a DC-present sample, only a subset of elements should correspond to the DC. The **Max Active Ratio (MaxAR)** term discourages LAPs from assigning high importance probabilities to irrelevant elements, such as background regions, by limiting the proportion of elements selected as relevant.

- **Exclusion:** In a DC-absent sample, no elements should be attributed to that DC. The **Inactive Ratio (IAR)** term discourages LAPs from assigning high importance probabilities to elements associated with irrelevant DCs.

Thus, during training, LAP learns that elements that frequently appear in samples associated with a particular DC, and are absent in others, represent the true evidence for that DC.

To express it formally, each DC, $dc_m \in DC^{1:M}$, corresponds to discriminative elements that favor $Y^{dc_m^+}$ over $Y^{dc_m^-}$. We leverage this definition to develop a weakly supervised training scheme for LAPs, enabling them to learn to detect these discriminative elements for each DC, without relying on expert annotations. As shown in Eq. (4), the supervision loss for the $l$-th LAP, denoted as $L_{LS_l}$, is composed of three terms for each DC $dc_m$. The **MinAR** and **MaxAR** loss terms are applied to DC-present samples (i.e., those with ground truth in $Y^{dc_m^+}$), while the **IAR** is applied to DC-absent samples (i.e., those with ground truth in $Y^{dc_m^-}$). The loss terms are normalized by the number of DC-present and DC-absent samples for $dc_m$, represented as $n_m^+$ and $n_m^-$, respectively. The 3 terms are explained in detail below.

$$L_{LS_l} = \sum_{m=1}^{M} \left( \frac{\sum_{(X^l,y):y \in Y^{dc_m^+}} \left( L_{LS_{l,m}}^{MinAR}(X^l,y) + L_{LS_{l,m}}^{MaxAR}(X^l,y) \right)}{n_m^+} + \frac{\sum_{(X^l,y):y \in Y^{dc_m^-}} L_{LS_{l,m}}^{IAR}(X^l,y)}{n_m^-} \right) \quad (4)$$

The **MinAR** loss encourages LAP to assign high importance probabilities to at least a specified portion of elements ($\gamma^{MinAR}$) in DC-present samples, ensuring sufficient coverage. Conversely, the **MaxAR** loss restricts LAP from assigning high importance to more than a particular portion of elements in DC-present samples ($\gamma^{MaxAR}$), promoting precise localization, especially in deeper layers with larger receptive fields. Since DC-related input regions may appear in the receptive fields of all elements, the network might otherwise consider all of them DC-related. Finally, the **IAR** loss penalizes high importance probabilities in DC-absent samples, compelling LAP to avoid false activations. To maintain balance with MinAR, IAR is applied only to a selected portion ($\gamma^{IAR}$) of challenging elements rather than the entire set. The loss terms for the $l$-th LAP and the $m$-th DC, given a sample with input embedding $X^l$ to the LAP and ground-truth label $y$, are computed as follows. Let $I^{l,m}$ denote the output of the $l$-th LAP for the $m$-th DC for input $X^l$ according to Eq. (3). The operators Top-K$(I^{l,m})$ and Bot-K$(I^{l,m})$ return the indices of the $K$ largest and smallest elements of $I^{l,m}$, respectively, and $n_e$ denotes the number of elements in $X^l$. Using these definitions, the corresponding loss terms are:

$$L_{LS_{l,m}}^{MinAR}(X^l,y) = \frac{-2}{K_{MinAR}} \times \sum_{i \in \text{Top-K}_{MinAR}(I^{l,m})} \ln(I_i^{l,m}) \quad ; \quad K_{MinAR} = \lceil \gamma^{MinAR} \times n_e^l \rceil$$

$$L_{LS_{l,m}}^{MaxAR}(X^l,y) = \frac{-1}{K_{MaxAR}} \times \sum_{i \in \text{Bot-K}_{MaxAR}(I^{l,m})} \ln(1 - I_i^{l,m}) \quad ; \quad K_{MaxAR} = \lceil \left(1 - \gamma^{MaxAR}\right) \times n_e^l \rceil \quad (5)$$

$$L_{LS_{l,m}}^{IAR}(X^l,y) = \frac{-1}{K_{IAR}} \times \sum_{i \in \text{Top-K}_{IAR}(I^{l,m})} \ln(1 - I_i^{l,m}) \quad ; \quad K_{IAR} = \lceil \gamma^{IAR} \times n_e^l \rceil$$

In equation 5, $I_i^{l,m}$ denotes the importance probability calculated for the $i$-th element for the $m$-th DC in $I^{l,m}$. For **MinAR**, positive cross-entropy loss is calculated over the $K_{\text{MinAR}}$ elements with the highest importance probabilities for the $\text{dc}_m$ ($i \in \text{Top-K}_{\text{MinAR}}(I^{l,m})$), whereas for **MaxAR**, negative cross-entropy loss is calculated over the $K_{\text{MaxAR}}$ elements with the lowest importance probabilities ($i \in \text{Bot-K}_{\text{MaxAR}}(I^{l,m})$). In contrast, for **IAR**, which is applied on the DC-absent samples, negative cross-entropy loss is calculated over $K_{\text{MinAR}}$ elements with the highest importance scores ($i \in \text{Top-K}_{\text{IAR}}(I^{l,m})$), i.e., the challenging ones that are mistakenly scored high. More notes on pixel selection are provided in App. C.5. Notably, the first term is multiplied by 2 to balance the effect of positive and negative losses for $\text{dc}_m$. $\gamma^{\text{MinAR}}$, $\gamma^{\text{MaxAR}}$, and $\gamma^{\text{IAR}}$ are hyper-parameters. If general knowledge about the domain exists, e.g., the expected size range of Distinguishing Clues (DCs), it can be injected into training through these parameters. Otherwise, they can be treated like standard hyper-parameters and selected based on validation performance. Further details are provided in App. D.1.1.

Although the training scheme is designed not to rely on expert annotations, if someone wishes to use detailed annotations, the proposed method can also fully inject expert knowledge into the training. The details are provided in App. C.6. Additionally, beyond the main loss terms, optional consistency losses can be applied to encourage agreement between output probability maps from different LAPs. Further details are provided in App. C.7.

## 3 Experiments

The primary purpose of the experiments is to assess the success of the proposed method, LAP, in interpreting models compared to other white-box local XAI methods, from the perspective of faithful localization. Comparisons were conducted under both post-hoc explanation and training-from-scratch schemes to evaluate general applicability. To this end, we compared LAP's interpretations with those from widely used white-box local XAI methods that require little to no modification, as well as recognized methods published in top-tier venues: Guided Back-propagation (GBP) (Springenberg et al., 2014), Grad-CAM (GC) (Selvaraju et al., 2017), Guided Grad-CAM (GGC) (Selvaraju et al., 2017), DeepLIFT (DL) (Shrikumar et al., 2017), and Relevance-CAM (RC) (Lee et al., 2021a), as well as one unsupervised concept-based local explanation method, CRAFT (CR) (Fel et al., 2023). Additionally, we compared LAP with the self-interpretability method BotCL (Wang et al., 2023), which requires network modification and tuning. Implementations from Captum (Kokhlikyan et al., 2020) in PyTorch (Ansel et al., 2024) were used for the first four methods, while the original GitHub repositories were used for the latter two. Furthermore, the performance and interpretability of the LAP-extended architectures were compared with those of the original architectures to assess LAP's impact on overall performance as a secondary output of this study.

In the experiments, LAP's generality was validated on two common CNN architectures, ResNet (He et al., 2016) and Inception-V3 (Szegedy et al., 2016). We evaluated our method on two datasets from different domains: a medical dataset and the well-known ImageNet. Further details regarding the choice of architectures and XAI methods for comparison are provided in App. C.8. The experimental setup and results for these datasets are presented in the following subsections.

**Baselines:** We consider Guided Back-propagation (GBP) (Springenberg et al., 2014), Grad-CAM (GC) (Selvaraju et al., 2017), Guided Grad-CAM (GGC) (Selvaraju et al., 2017), DeepLIFT (DL) (Shrikumar et al., 2017), Relevance-CAM (RC) (Lee et al., 2021a), CRAFT (CR) (Fel et al., 2023), and BotCL (BCL) (Wang et al., 2023) as baselines in our experiments.

**Datasets:** We consider datasets, RSNA Pneumonia Detection (Anouk Stein et al., 2018), and ImageNet Object Recognition (Russakovsky et al., 2015).

### 3.1 Experimental Setup Details

We evaluated three configurations of LAP adoption across the two datasets: (1) Training detached LAPs to explain frozen pretrained networks; (2) Training LAP-extended architectures from scratch, initialized with random weights matching the original backbone; (3) Fine-tuning LAP-extended architectures initialized with the pretrained backbone and the trained detached LAPs, for a small number of epochs. In all settings,

we optimized the trainable components jointly using the Adam optimizer (Kingma & Ba, 2014) with a learning rate of 1E-4 and a weight decay of 1E-6. We also applied the same data augmentations as those used for training the original network. For the two configurations involving training or tuning LAP-extended architectures, we incorporated the additional LAP-supervision loss alongside the primary task loss. The weighting factor of the LAP-supervision term was selected via hyperparameter tuning to optimize performance metric on the main task, i.e. classification. This weight was fixed after selection over one of the architectures and applied uniformly across all architectures. Models were trained until convergence, and the checkpoint achieving the best validation performance on the primary task was retained. For detached LAPs, the checkpoint with the lowest validation loss was selected. Dataset-specific training details are provided in the corresponding subsections.

During the hyper-parameter configuration phase, we evaluated various strategies for encouraging consistency across the DC-wise importance maps produced by LAPs at different depths of the network. We found that using the sum of the DC-wise score maps from all LAPs as reference signals within the MinAR and MaxAR terms of the LAP-Supervision loss yielded the most robust and effective form of inter-LAP consistency, leading to improved interpretability across datasets. This configuration showed stable performance in practice and was therefore adopted for all experiments in the paper. In preliminary experiments, we also evaluated auxiliary consistency losses in addition to the LAP-Supervision objective. However, their effect on performance and interpretability was negligible compared to the consistency already induced by LAP-Supervision, so we do not discuss them further in the main experimental analysis.

### 3.2 RSNA pneumonia detection

**Experimental Setup:** The RSNA pneumonia dataset was published in a Kaggle challenge in 2018 (Anouk Stein et al., 2018). The dataset contains chest X-ray images of 8851 healthy people and 6012 patients with lung pneumonia. Pneumonia regions are annotated by experts using bounding boxes. We used the dataset for a binary classification task with two base architectures: ResNet-18 and Inception-V3. To assess the effect of LAPs as internal modules, we also adopted LAP-extended ResNet-18 and LAP-extended Inception-V3 architectures, in each of which 4 LAP-Pools were used in different layers from shallow to deep ones (App. D.1.2). According to Sec. 2.1, we used a single DC, "pneumonia," to interpret the decisions. We trained LAP-extended models using two approaches: weak supervision without expert annotations (WSL-), following the explanations in Sec. 2.3, and experts' knowledge injection (BBL-), where experts' bounding boxes were used to train the LAPs as described in App. C.6. Additionally, we applied the self-interpretation method of BotCL to train BotCL-extended architectures (BCL-) (Wang et al., 2023), which served as another baseline for comparison.

The experiments were repeated five times with different seeds to report the mean and standard deviation.

**Classification Performance:** We used two metrics to evaluate model classification performance on the test set: Balanced Accuracy (BA) and F1-Score. The results are reported for all four versions of each of the two architectures in ResNet-18 and Inception-V3. According to the results presented in Tab. 1, not only LAP-extension has not resulted in performance degradation in any of the architectures, but the average performance of the LAP-extended models slightly surpasses that of the vanilla (p-value $\leq 0.05$ in an unpaired t-test) and BotCL-extended models. Notably, the performance of the LAP-extended model trained with the proposed weak supervision method (**WSL-**), which does not rely on expert annotations, closely aligns with that of the LAP-extended model trained using expert bounding boxes (**BBL-**). Detailed performance metrics are provided in App. D.2.1.

**Evaluation of XAI methods:** Several evaluation metrics have been introduced in the literature for evaluating XAI methods. In this study, we have employed two distinct evaluation methods for our two datasets based on the nature of their domain (further explanations are provided in App. C.8.1). For the RSNA dataset, following the "object localization" method (Lee et al., 2021a), the Intersection over Union (IoU) metric was calculated between the binarized attribution maps and the bounding boxes provided by experts for correctly predicted positive cases of pneumonia. Various binarization approaches have been adopted in the literature, ranging from selecting a fixed portion, e.g., 15% of the top-scored pixels (Selvaraju et al., 2017), to adopting a threshold of mean + standard deviation ("$\mu + \sigma$") over the individual attribution

Table 1: Classification F1-Score and Balanced Accuracy (BA) on the RSNA test dataset for the vanilla ResNet18 and Inception3 models, BotCL-extended models (BCL-), LAP-extended models trained using the proposed weak supervision (WSL-), and LAP-extended models trained using expert annotations (BBL-). The results demonstrate that not only LAP-extension does not result in performance degradation, but LAP-extended models achieve slightly higher average performance metrics compared to the vanilla and BotCL-extended models across both architectures. Notably, the WSL version trained without expert annotations performs on par with the BBL version trained with expert annotations. All values are reported as mean $\pm$ standard deviation over five independently trained models initialized with different random seeds.

| | ResNet 18 (RN) | | | | Inception V3 (I3) | | | |
|---|---|---|---|---|---|---|---|---|
| | RN | BCL-RN | WSL-RN | BBL-RN | I3 | BCL-I3 | WSL-I3 | BBL-I3 |
| F1 | 94.58±0.27 | 94.79±0.26 | 94.88±0.22 | **94.92**±0.19 | 95.15±0.41 | 95.22±0.25 | 95.28±0.19 | **95.32**±0.24 |
| BA | 94.52±0.25 | 94.74±0.26 | 94.72±0.27 | **94.86**±0.2 | 95.14±0.39 | 95.17±0.28 | **95.25**±0.17 | 95.21±0.3 |

maps (Lee et al., 2021a). We used three approaches to assess attribution maps from different perspectives. Note that the BotCL interpretation method applies only to BotCL-extended architectures. The results for "$\mu + \sigma$" are presented in Tab. 2, and the results for the other approaches are available in App. D.2.2. Notably, LAPs inherently provide a classification over the attribution maps and do not require external thresholds. We adopted LAP as a post-hoc explainer for the vanilla and the BotCL-extended architectures, and as an intrinsic interpreter for the LAP-extended ones. Across all three binarization approaches, the IoU for LAPs is significantly higher than that of the other XAI methods for all models (p-value $\leq 0.001$ in a paired t-test).

The IoU of LAP's importance maps with the ground truth bounding boxes ranges from 24% to 43%. In comparison, the next most successful methods, Relevance-CAM (RC) and CRAFT, range from 7% to 36% and 4% to 35%, respectively, across the same models. Additionally, LAPs trained with the proposed weak-supervision method (requiring no expert annotations) achieve 35-36% IoU, compared to 41-43% IoU for LAPs trained with expert bounding boxes.

Interestingly, the attribution maps of CRAFT and Relevance-CAM, which outperform the other methods except for LAP, also improve in LAP-extended architectures (32-35% and 26-36%, respectively), compared to the vanilla architectures (21-27% and 7-28%), while decreasing in BotCL-extended architectures (4-18% and 7-20%).

Although perturbation-based evaluations such as pixel flipping can introduce severe out-of-distribution image effects in medical imaging, we additionally compute SRG scores (Bluecher et al., 2024) to assess the faithfulness of the XAI methods on the RSNA dataset. The results are presented and discussed in detail in App. D.2.3. Remarkably, the findings are fully consistent with the IoU-based assessment, with LAP achieving the highest scores across all architectures. LAP's SRG scores range from 0.23 to 0.47, whereas Relevance-CAM and CRAFT range from 0.01–0.40 and 0.02–0.40, respectively. The LAP-based training scheme, specially the version that does not rely on expert annotations, also improves the SRG scores obtained by LAP itself (by an average of 0.11). Notably, it also enhances SRG scores for Relevance-CAM (by an average of 0.39) and CRAFT (by an average of 0.22), indicating that the proposed approach increases the model's inherent interpretability. Interestingly, even when used as detached post-hoc explainers for the self-interpretable BotCL-extended architectures, LAP achieves higher SRG scores than BotCL's own native interpretations (an average improvement of 0.18), further demonstrating the faithfulness of the approach despite detached LAPs not being part of the model's decision-making.

**Model Diagnosis via Attribution Maps:** LAPs also enable us to diagnose the decision-making process for each individual image by providing through-the-depth interpretations. Interpretations of LAPs for four examples are shown in Fig. 3 using a WSL-ResNet-50 model. The sources of errors made by the model in the first and third cases are identifiable. In the first case, the initial two LAPs correctly detected the pneumonia region, but the model later altered its decision. In the third case, the model incorrectly identified pneumonia regions highlighted by the LAPs. Notably, the final LAP interpretation is aligned with the model's decisions. It can also be observed that the aggregated result, denoted as "LAP All," is as high-resolution as the first LAP while being as accurate as the last one. For true-positive predictions, LAP's provided 's provided attribution maps show a satisfactory overlap with infection bounding boxes, whereas other methods fail to capture all

Table 2: IoU of the binarized attribution maps with expert annotations on correctly classified positive RSNA test samples. Results are provided for the vanilla ResNet18 and Inception3 models, BotCL-extended models (BCL-), LAP-extended models trained using the proposed weak supervision (WSL-), and LAP-extended models trained using expert annotations (BBL-). LAP-based interpretations produce binarized maps that are significantly closer to expert annotations compared to other XAI methods across all eight architectures. Additionally, while the BotCL approach reduces similarity to expert annotations relative to the vanilla models, the proposed LAP supervision loss enhances the detection of relevant regions, even for other XAI methods such as CRAFT and Relevance-CAM. All values are reported as mean ± standard deviation over five independently trained models initialized with different random seeds. Not Applicable (NA): The BotCL interpretation method applies only to BotCL-extended architectures.

| Models | IoU Between Expert Annotations and Binarized Attribution Maps (Threshold: $\mu + \sigma$) | | | | | | | |
|---|---|---|---|---|---|---|---|---|
| | DL | GGC | GC | GBP | RC | CR | BotCL | LAP |
| ResNet18 | 13.6±0.81 | 8.99±1.46 | 22.63±1.47 | 7.86±1.4 | 6.9±1.49 | 21.29±1.89 | NA | **31.37**±2.79 |
| BCL-RN | 14.04±1.39 | 8.21±2.10 | 17.05±1.58 | 9.56±0.79 | 20.05±4.37 | 17.79±4.22 | 17.19±3.52 | **25.01**±2.48 |
| WSL-RN | 15.07±0.51 | 12.21±0.77 | 13.85±1.75 | 12.56±0.77 | 34.24±0.99 | 32.41±1.41 | NA | **36.28**±1.19 |
| BBL-RN | 15.72±0.67 | 13.95±0.56 | 18.87±5.03 | 13.77±0.91 | 25.75±3.81 | 34.92±5.28 | NA | **43.26**±1.51 |
| Inception3 | 13.12±0.73 | 7.73±0.78 | 26.86±0.96 | 8.17±0.81 | 27.93±0.97 | 26.86±1.30 | NA | **29.52**±0.93 |
| BCL-I3 | 12.66±0.68 | 5.68±2.50 | 3.10±3.59 | 7.81±0.62 | 7.03±4.87 | 3.96±5.60 | 1.62±1.22 | **24.40**±2.05 |
| WSL-I3 | 14.14±0.51 | 6.09±1.19 | 7.89±2.14 | 6.82±1.62 | 33.43±0.76 | 32.87±1.01 | NA | **35.75**±0.62 |
| BBL-I3 | 15.28±0.41 | 8.12±1.54 | 21.03±3.36 | 8.43±1.67 | 36.35±1.13 | 31.77±3.32 | NA | **41.46**±1.87 |

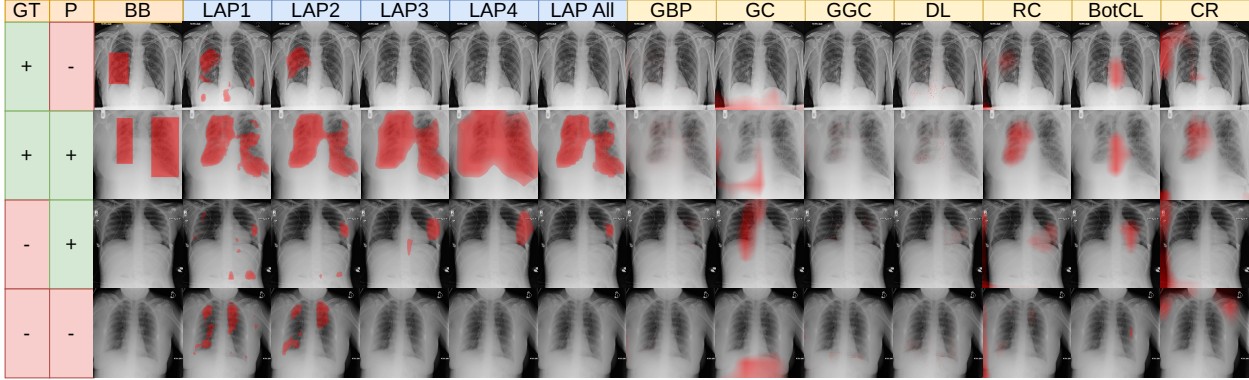

Figure 3: Examples of attribution maps provided for the WSL-ResNet18 model by LAPs at different network depths, the aggregated LAP, and other XAI methods, compared to RSNA bounding boxes (BB). LAP is faithful to the model's prediction (P), showing from which layer the model has differed from the ground truth (GT).

boxes. Overall, the regions highlighted by LAP are also more reasonable. Additional examples from other models are provided in App. D.6. Each LAP can also function as a standalone predictor, aiding in model diagnosis. Related experiments are detailed in App. D.5.1.

**Ablation Studies:** To assess the effectiveness of the proposed weakly supervised training scheme, we trained the LAP-extended ResNet-18 architecture under three additional settings: two naive weak-supervision methods, in which the MaxAR loss term is omitted and IAR is applied to all pixels ($\gamma^{\mathrm{IAR}} = 1$); in one of these, the MinAR term is applied to only the top pixel ($\gamma^{\mathrm{MinAR}} = \frac{1}{HW}$), while in the other, MinAR is applied to all pixels ($\gamma^{\mathrm{MinAR}} = 1$). For the third setting, a hybrid approach is adopted, in which MinAR is applied to a proportion of pixels according to the proposed method ($\gamma^{\mathrm{MinAR}} = 0.1$), while IAR and MaxAR terms remain the same as in the naive weak-supervision settings. The classification performance metrics and the IoU of the binarized attribution maps with expert bounding boxes are reported in Tab. 3.

It is observed that the proposed training scheme leads to the highest self-interpretability (36.28%) while maintaining task-specific performance close to the best configuration (F1 of 94.88% vs. 95.00%). In contrast,

configurations with naive settings exhibit significant degradation in interpretability (13.94%). Additionally, we conducted another ablation study to assess the sensitivity of results to the ratio hyper-parameters of the LAP Supervision Loss. The results presented in App. D.2.5 confirm the suitability of the proposed approach for setting these hyper-parameters based on a general knowledge of the problem domain (App. D.1.1). According to the results, the ratio of $\gamma^{\mathrm{MinAR}}$ is the most influential one; when it is large (0.5) or small (0.01), it resembles the naive weakly supervised settings, and the results degrade (IoU of 25% vs. 36%). The most suitable value for this parameter corresponds to the expected distinguishing clue size (0.1 in this problem).

Table 3: Comparison of classification performance (Balanced Accuracy (BA) and F1-Score) and similarity of binarized attribution maps to expert bounding boxes (IoU) for LAP-extended ResNet-18 models. Models were trained under four different settings: the proposed weak-supervision scheme, two naive weak-supervision settings, and one hybrid setting. The model trained with the proposed scheme produces attribution maps closest to the expert annotations while maintaining classification performance near the best configuration. In contrast, models trained with the naive settings exhibit substantial degradation in their attribution maps. All values are reported as mean ± standard deviation over five independently trained models initialized with different random seeds.

| Configuration | BA | F1 | IoU |
|---|---|---|---|
| Proposed method ($\gamma^{\mathrm{MinAR}} = 0.1, \gamma^{\mathrm{MaxAR}} = 0.5, \gamma^{\mathrm{IAR}} = 0.1$) | 94.72±0.27 | 94.88±0.22 | **36.28**±1.19 |
| Hybrid method ($\gamma^{\mathrm{MinAR}} = 0.1, \gamma^{\mathrm{MaxAR}} = 1, \gamma^{\mathrm{IAR}} = 1$) | 94.79±0.22 | 94.87±0.21 | 31.90±1.35 |
| Naive weak-supervision V1 ($\gamma^{\mathrm{MinAR}} = \frac{1}{HW}, \gamma^{\mathrm{MaxAR}} = 1, \gamma^{\mathrm{IAR}} = 1$) | **94.92**±0.22 | **95.00**±0.20 | 4.40±1.23 |
| Naive weak-supervision V2 ($\gamma^{\mathrm{MinAR}} = 1, \gamma^{\mathrm{MaxAR}} = 1, \gamma^{\mathrm{IAR}} = 1$) | 94.61±0.25 | 94.74±0.19 | 13.94±0.08 |

### 3.3 Imagenet

**Experimental Setup:** In this experiment, we examined the applicability of the proposed method to already trained models by evaluating its performance on the ImageNet classification task (Russakovsky et al., 2015), which involves recognizing objects with substantial variation in scale. We employed three architectures, ResNet50, InceptionV3, and EfficientNetB0, from the torchvision model zoo (maintainers & contributors, 2016). Each model was evaluated under two paradigms: post-hoc explanation and plugged-in interpretation. Post-hoc explanation uses detached LAPs, which are trained while the backbone network is frozen and therefore do not affect the network's classification performance. In contrast, plugged-in LAPs involve fine-tuning the full LAP-extended architecture for a few epochs. For each architecture, two LAPs were implemented, with concept sets defined using the one-vs-others strategy described in Sec. 2.1 to interpret each class. Additional details regarding the experimental setup are provided in App. D.1.3.

**Classification Performance:** The performance of the base (original) models was first assessed using the Top-1 accuracy metric and then compared with their LAP-extended counterparts, in which the LAP module was plugged into the pretrained networks and tuned for a few epochs. The LAP extension yielded a slight improvement in ResNet50 and InceptionV3 architectures (from 76.14% to 76.40% and 76.08% to 76.16% respectively), and resulted in no change for EfficientNetB0, which achieved 77.67% accuracy.

**Model Diagnosis via Attribution Maps:** Examples of LAP interpretations are provided in Fig. 4, highlighting which input regions resulted in each of the top-3 predictions. For example, in an image containing an airship and cars, the yellow coloration near the cars inadvertently increased the prediction score for the taxi class. Additional examples are available in App. D.6.

**Evaluation of XAI methods:** For comparing LAP's interpretation with other XAI methods, we followed the method presented in (Petsiuk et al., 2018) (the reasons are discussed in App. C.8.1). The XAI methods were evaluated across two target layers: the last convolutional layer, after any subsequent normalization and activation layers, commonly utilized by various XAI methods like CAM. This layer, while offering valuable insights, has low resolution and does not allow assessment of intermediate layers; and another layer before the last non-adaptive pooling that has higher resolution.

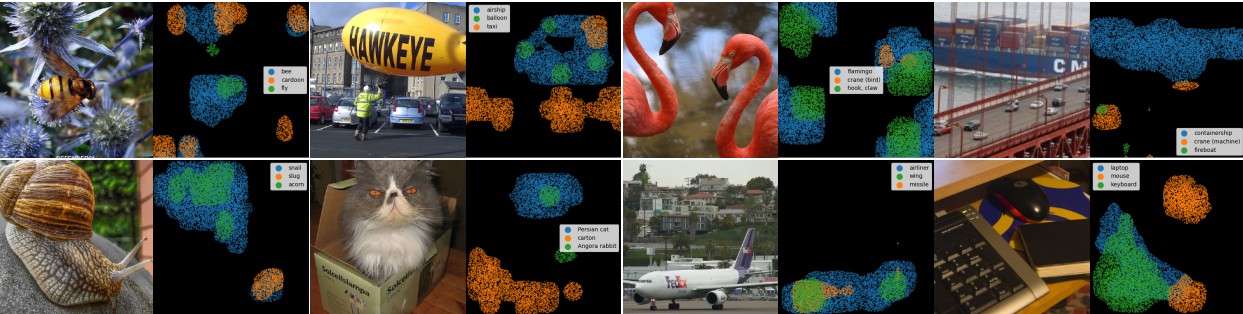

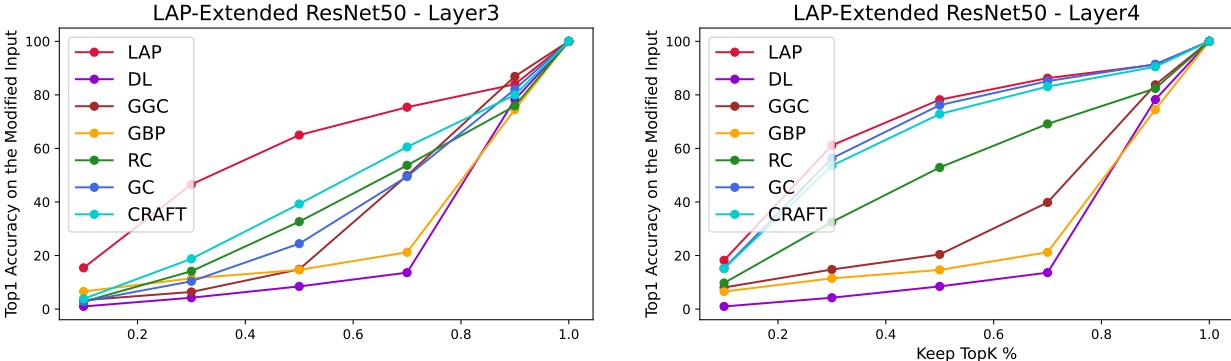

Figure 4: Interpretations of LAP-ResNet50 on ImageNet, highlighting the regions corresponding to each of the top-3 predicted classes as indicated in the legend.

We compared the faithfulness of different XAI methods on each model by modifying the input images based on their provided attribution maps. Faithfulness was evaluated by keeping $k\%$ pixels with the highest scores provided by the XAI method, zeroing out the rest of the image, and then passing the modified image through the network for re-evaluation. The results for LAP-extended ResNet50 are presented in Fig. 5 for percentages of 10%, 30%, 50%, 70%, and 90%. In this experiment, the attribution maps were calculated based on the predicted class, and the results were evaluated with respect to the original predictions. A faithful XAI method would capture the influencing regions for decision-making in the model, thereby keeping the original decision for the modified image.

Across all percentages, LAP retains higher original prediction accuracy than the other XAI methods, with the effect most pronounced for the top 10-30% of pixels. Additionally, while the faithfulness of other methods decreases notably when interpreting layer 3 as the target, LAP maintains its performance.

The results for other architectures can be found in App. D.3.1, showing similar trends. LAP overperforms all methods in all 12 cases, but two which is slightly overperformed by GradCam.

Figure 5: Assessing the faithfulness of different XAI methods on the ImageNet dataset for Lap-Extended ResNet 50 over the outputs of layers 3 and 4. Faithfulness is evaluated by keeping a ratio of pixels ranked by the XAI method (K%), zeroing out the rest, and evaluating the prediction on the modified image w.r.t the prediction of the original image. LAP is shown to be more faithful than the other models, especially when interpretation is demanded for the middle layers.

Ultimately, to reduce sensitivity to out-of-distribution samples and eliminate ranking inconsistencies across different faithfulness assessment configurations, we adopted the SRG score introduced in (Bluecher et al., 2024). This score measures the area between the accuracy curves generated by pixel-flipping (PF) under two strategies: eliminating least important pixels first (LIF) and most important pixels first (MIF). A larger area

indicates a more faithful XAI method. Results for the original and LAP-extended versions of ResNet-50, Inception V3, and EfficientNet B0 are presented in Tab. 4 for two target layers: the last convolutional block and an intermediate convolutional layer. According to the results, LAP outperforms the other methods in 9 out of 12 configurations and is only slightly surpassed in the remaining three by either Relevance-CAM or Grad-CAM. Notably, LAP achieves a noticeably higher average score across all baselines (0.48 versus 0.42 for Relevance-CAM, the second-best method), with the performance gap being particularly pronounced when interpreting middle layers as the target. As is evident for all baseline methods, interpretations and explanations targeting the last convolutional block yield substantially higher scores than those targeting middle layers. To ensure a strictly fair comparison under this shared evaluation behavior, our reported numbers use only the LAP interpretation map at the target layer itself, without applying any cross-layer aggregation. This isolates the performance of a single LAP in that layer and places it on equal footing with how the baselines are evaluated.

Table 4: SRG scores of XAI methods on original and LAP-extended versions of ResNet50, Inception V3, and EfficientNet B0. Scores are calculated for interpretations from two target layers: the last convolutional layer (LC) and an intermediate layer. LAP outperforms the other methods in 9 out of 12 configurations, is only slightly surpassed in the remaining three, and achieves a noticeably higher average score compared to the baselines.

| Model (Target) | DL | GGC | GC | GBP | RC | CR | LAP |
|---|---|---|---|---|---|---|---|
| ResNet50 (Layer3) | 0.14 | 0.21 | 0.16 | 0.16 | 0.36 | NC [*] | **0.42** |
| ResNet50 (LC) | 0.14 | 0.21 | 0.43 | 0.16 | 0.44 | NC | **0.45** |
| LAP-ResNet50 (Layer3) | 0.14 | 0.19 | 0.16 | 0.16 | 0.37 | 0.21 | **0.44** |
| LAP-ResNet50 (LC) | 0.14 | 0.22 | 0.45 | 0.16 | 0.47 | 0.41 | **0.48** |
| InceptionV3 (Mixed7a) | 0.27 | 0.22 | 0.32 | 0.19 | 0.40 | NC | **0.56** |
| InceptionV3 (LC) | 0.27 | 0.29 | 0.57 | 0.19 | 0.56 | NC | **0.58** |
| LAP-InceptionV3 (Mixed7a) | 0.22 | 0.23 | 0.34 | 0.16 | 0.41 | NC | **0.56** |
| LAP-InceptionV3 (LC) | 0.22 | 0.28 | 0.53 | 0.16 | **0.56** | NC | 0.55 |
| EfficientNetB0 (Layer6) | 0.10 | 0.12 | 0.15 | 0.11 | 0.3 | NC | **0.41** |
| EfficientNetB0 (LC) | 0.10 | 0.16 | **0.47** | 0.11 | 0.44 | NC | 0.43 |
| LAP-EfficientNetB0 (Layer6) | 0.10 | 0.12 | 0.15 | 0.11 | 0.3 | 0.15 | **0.41** |
| LAP-EfficientNetB0 (LC) | 0.10 | 0.16 | **0.47** | 0.11 | 0.44 | NC | 0.42 |
| Average | 0.16±0.06 | 0.20±0.05 | 0.35±0.15 | 0.15±0.03 | 0.42±0.08 | NC | **0.48±0.06** |

[*] Not Calculated: Due to heavy computational costs, the CRAFT method (CR) was only applied to 3 configurations.

## 3.4 Computational Cost Analysis

In our experiments, LAP was adopted in three settings: (1) training LAP-extended architectures from scratch; (2) fine-tuning LAP-extended pretrained architectures for a few epochs (up to five); (3) training detached LAPs as post-hoc explainers for a frozen model. All LAP-based interpretation approaches require some level of training; however, the post-hoc variant is the lightest, as it only trains the detached LAP heads for a few epochs. Detailed measurements of training time are provided in App. D.4. According to the results, training LAP-extended models increases total training time by approximately a factor of 1.5 compared to training the original architecture, mainly due to the additional LAP-supervision losses and the computation of Top-K elements. In contrast, the time required to train the detached LAPs is comparable to (or lower than) that of training the original network for one epoch. Since detached LAPs typically require at most five epochs on ImageNet to converge when the main model is frozen, this makes the cost of providing LAP-based interpretation roughly 2.5% of training the original network from scratch.

Among baseline methods, CRAFT also requires learning concepts for post-hoc explanation. Running the official GitHub implementation took approximately 5-10 days for the ImageNet dataset, depending on the backbone architecture, as reported in App. D.4. In contrast, BotCL, which embeds self-interpretation into the architecture, must be trained from scratch and is not applicable to pretrained architectures. This makes BotCL impractical for large datasets like ImageNet. Other baselines, including Grad-CAM, Guided Grad-CAM, GuidedBackprop, DeepLIFT, and Relevance-CAM, require no additional training, making them trivially applicable to any pretrained model.

We also measured the runtime impact of performing interpretation during inference. Full inference-plus-interpretation times for all methods and architectures are reported in App. D.4. On the RSNA dataset, LAP required 2.79 seconds on average for ResNet-18 and its variants and 5.72 seconds on Inception-V3 and its variants for processing the entire test set on a Tesla T4 GPU. On ImageNet, LAP required an average of 185 seconds across all architectures to process the complete validation set on a Quadro 8000 GPU. These values compare favorably with Grad-CAM, one of the fastest baselines, which required 3.27 s, 34.22 s, and 138 s in the corresponding settings, indicating that LAP remains computationally efficient and practical in the inference time.

## 4 Discussion

In this section, we provide the discussion of our results in previous section. Interpretations were produced by training a module called LAP using our designed weakly supervised loss, without relying on expert annotations. The module supported two paradigms: post-hoc and plugged-in. In the post-hoc paradigm, LAP modules were trained separately while the base network remained frozen, ensuring that the main model's performance was unaffected. In the plugged-in paradigm, LAP modules were integrated into the model architecture. The extended architecture was either trained from scratch using both the LAP supervision loss and the task-specific loss, or fine-tuned on pretrained networks for a few epochs to learn interpretations. We assessed interpretations across two datasets from different perspectives.

The attribution maps provided by different XAI methods in the RSNA dataset were assessed on correctly classified positive test samples by comparing them to the expert bounding boxes using the IoU metric, which represent the expected and clinically relevant regions. We observed five key findings:

1. Attribution maps generated by LAP were significantly closer to the expert bounding boxes across all four extensions of the two base architectures (p-value $\leq 0.001$).

2. LAP-extended architectures trained using the proposed weak-supervision method (requiring no expert annotations) achieved interpretation quality comparable to models trained with expert bounding boxes (35-36% IoU compared to 41-43%). This highlights the practical value of the method, as acquiring pixel-level expert labels is expensive and often infeasible, especially for large-scale datasets.

3. Among the three XAI methods with the highest average IoU scores, i.e. LAP, Relevance-CAM, and CRAFT, LAP was the only method that did not exhibit significant degradation across any architecture. In contrast, Relevance-CAM and CRAFT degraded to IoU values as low as 3.96% and 7.05%, respectively, whereas the lowest IoU observed for LAP remained at 24%, indicating that competing methods are less consistent and may fail on certain model configurations.

4. The performance of all three methods degraded in architectures extended and trained using the BotCL self-interpretation method compared to the original architectures (from ranges of 30-31%, 7-28%, and 21-27% to 24-25%, 7-20%, and 14-18%, respectively). In contrast, all methods improved in both LAP-extended architectures, including the model trained without expert bounding boxes as supervision (36-36%, 33-34%, and 32-33%, respectively). These results suggest that LAP-based supervision during training enhances overall model interpretability, benefiting not only LAP but also other XAI methods.

   The SRG-based results were fully consistent with the IoU-based assessment of XAI methods. LAP achieved the highest scores across all architectures (0.23–0.47, compared to 0.01–0.40 for Relevance-CAM, the strongest competing baseline). The proposed weakly supervised training scheme further improved SRG scores not only for LAP but also for Relevance-CAM and CRAFT (average increases of +0.11, +0.39, and +0.22, respectively). Moreover, even when used as detached LAPs for post-hoc explanation of the self-interpretable BotCL-extended architectures, LAP obtained SRG scores that were on average 0.18 higher than BotCL's own interpretations, indicating its faithfulness to the model's internal decision-making process.

5. We further demonstrated the diagnostic value of binarized attribution maps by visualizing the maps provided by LAP and other XAI methods on four representative cases with varying ground-truth and

predicted labels. LAP consistently identified regions that were more clinically plausible for model decision-making and produced binarized regions of appropriate size across images with differing expert bounding box scales, unlike competing methods. This property was particularly valuable when attribution maps were used as auxiliary outputs, such as for treatment planning in medical applications. Moreover, the layered architecture of LAP-based interpretation enabled analysis of which network components contributed to prediction errors, offering a useful tool for debugging and refining model architectures.

In the ImageNet dataset, the faithfulness of different XAI methods was evaluated by retaining varying percentages of the highest-scoring pixels, masking the remaining pixels, feeding the modified inputs to the model, and measuring the accuracy of reproducing the original predictions. This experiment evaluated interpretations using two different target layers, allowing assessment of each method's ability to provide in-depth, layer-wise interpretability, which is useful for model diagnosis. For both the original and LAP-extended versions of the three architectures, ResNet-50, Inception-V3, and EfficientNet-B0, LAP achieved the highest faithfulness in 10 out of 12 cases, and was marginally outperformed by Grad-CAM in the remaining two cases. Notably, LAP achieved strong accuracy when retaining only the top 10-30% of pixels, indicating precise localization of regions that drive model decision-making. Additionally, while all methods showed substantial degradation when a middle layer was used as the target, LAP maintained relatively consistent performance, highlighting its effectiveness for diagnosing model behavior across network depth. Ultimately, to have a faithfulness assessment less sensitive to out-of-distribution samples and the assessment configurations, the faithfulness of XAI methods were also assessed by calculating SRG scores accross all 12 configurations. LAP achieved a noticeably higher average SRG score compared to the other methods (0.48 vs. Relevance-CAM achieving 0.42 as the second-best method), while achieving the highest score in 9 out of 12 configurations.

Alongside the quality assessment experiments for XAI methods on both datasets, we also reported the computational cost of training detached LAPs as post-hoc explainers and LAP-extended models. Training LAP-extended architectures from scratch required approximately 1.5 times the training time of the original network, which is a reasonable cost given the substantial improvements in interpretability, not only for LAP itself, but also for other methods such as Relevance-CAM and CRAFT (yielding 5%-27% higher IoU). Moreover, on large-scale datasets like ImageNet, we showed that fine-tuning LAP-extended architectures or training detached LAPs requires only about 2.5% of the time needed to train the base network from scratch. This overhead is also negligible compared to CRAFT as a post-hoc explanation method, which require substantial training for post-hoc explanation (hours versus weeks). Although LAP is not completely training-free like Relevance-CAM, it offers additional benefits by improving interpretability specially across intermediate layers. Finally, we demonstrated that inference-plus-interpretation runtime is comparable to the fastest baselines, such as Grad-CAM, making LAP one of the most computationally efficient options at the inference time.

As an auxiliary experiment, we evaluated whether training LAP-extended models from scratch affects downstream performance. We found no evidence of performance degradation; instead, results showed a slight improvement. On the RSNA dataset, for both ResNet-18 and Inception-V3 architectures, the mean performance metrics improved relative to their vanilla counterparts (F1-scores increased from 94.58% and 95.15% to 94.88% and 95.28% without expert annotations, and further to 94.92% and 95.32% when expert annotations were used in training LAPs; p-value $\leq 0.05$, unpaired t-tests). We also observed that fine-tuning LAP-extended architectures on ImageNet produced small performance gains in two of the three models, with the largest improvement in LAP-extended ResNet-50, where accuracy increased from 76.16% to 76.40%. Despite these encouraging results, the ability of LAP-extension to enhance the performance of base architectures requires further systematic investigation, which we propose for future studies, as it falls outside the primary focus of this paper.

## 5 Related works

**Introduction & Gradient-Based Methods:** This study belongs to the category of white-box local XAI methods, where the method has full access to the model's architecture and intermediate feature maps. One of the earliest approaches in this domain is Class Activation Mapping (CAM) (Zhou et al., 2016), which

uses the weights of the fully connected layer to determine the importance of features for each class and then calculates pixel scores based on channel-wise activations and their corresponding importance, making it applicable only to networks with a single fully connected layer following the last convolutional layer. Gradient-based methods estimate the network's behavior using the gradients of class scores with respect to the input, approximating the network via a first-order Taylor expansion. In this view, the gradients serve as weights indicating feature importance. Vanilla Gradient (Simonyan et al., 2013) uses raw gradients to identify important features but produces noisy importance maps. Guided Backpropagation (Springenberg et al., 2014) filters out negative gradient flows to highlight important pixels, although this filtering can lead to false positives. Grad-CAM (Selvaraju et al., 2017) follows the same principle as CAM but uses gradients to determine channel importance, making it more broadly applicable. Guided Grad-CAM (Selvaraju et al., 2017) combines Guided Backpropagation and Grad-CAM to produce fine-grained importance maps.

**Score-Based Methods:**Gradient-based methods suffer from gradient saturation problems, leading to near-zero importance scores. Score-based methods like Layer-wise Relevance Propagation (LRP) (Bach et al., 2015; Binder et al., 2016) and DeepLift (Shrikumar et al., 2017) propagate scores instead of gradients to calculate the importance of neurons. LRP has defined layer-specific rules to distribute the relevance score of each neuron in each layer to their input neurons. The rules assign relevances based on the contribution of the input neurons to the neurons in the next layer. DeepLift uses a similar procedure to LRP but calculates scores based on the difference in the output achieved compared to a baseline input. However, defining a suitable and meaningful baseline for all applications is challenging. Recently, some works have combined the CAM with score-based XAI methods to improve both. Score-CAM (Wang et al., 2020) adopts a DeepLift-style scoring scheme to find the channel-wise increase of confidence and then uses the confidence scores in the CAM method. Similar to Score-CAM, Relevance-CAM (Lee et al., 2021a) uses the LRP method to find the importance of channels in any layer and then applies the CAM method to find the corresponding regions in the input. Relevance-CAM has demonstrated superior performance over other CAM-based methods.

**Concept-Based Methods:** Despite previous literature providing interpretations as attribution maps on raw features, concept-based XAI methods offer explanations in a more human-comprehensible manner by utilizing concepts as semantically meaningful groups of input features (Schwalbe, 2022). Initial concept-based methods depended on expert annotations (Kim et al., 2018), which are expensive and not available for all datasets. To address this limitation, later methods focused on mining concepts using unsupervised techniques such as clustering (Ghorbani et al., 2019; Kowal et al., 2024), matrix factorization (Zhang et al., 2021; Fel et al., 2023), and principal component analysis (Zhang et al., 2021). Alternatively, some approaches incorporate encoder-decoder modules within the main architecture to reconstruct embeddings while encoding a dense set of concepts (Akpudo et al., 2025). Recent studies on unsupervised local concept-based XAI methods enable localization of concepts within input data, either by extending conventional local XAI methods like LRP (Achtibat et al., 2023), GradCam (Fel et al., 2023), and unsupervised segmentations (Wang et al., 2024), or by calculating similarity with concept or prototype vectors (Chen et al., 2019; Wang et al., 2023; Huy et al., 2025; Zhu et al., 2025), or with CLIP's provided embeddings for the concepts' terms (Benou & Raviv, 2025; Zhang et al., 2025). Methods such as CRAFT (Fel et al., 2023) and MCPNet (Wang et al., 2024) have proposed techniques to identify concepts recursively in different network sections, rather than only in the layer before the final decision.

In summary, despite recent advances in expressing concepts in human-comprehensible ways, faithful and precise detection of the input regions influencing a model's decision-making remains a valuable yet underexplored challenge, as much of the recent focus has shifted toward concept definition and human understandability. Our method, as a novel local XAI method, focuses on faithful localization of these influential regions and, from this perspective, is comparable to both traditional raw-feature-based and recent concept-based local XAI methods.

In addition to the literature discussed in this chapter, a more extensive discussion of broader families of related work is provided in App. D.7.

# 6 Conclusion and Future Works

In this paper, we introduced a versatile framework for achieving local interpretations of deep neural networks with higher fidelity by leveraging a novel definition: Distinguishing Clues (DCs). Our framework comprised a Complete System of DCs that identified input regions promoting each network output, a module named LAP that localized these clues, and a novel training scheme that enabled LAPs to learn from general problem knowledge without relying on expert annotations.

We assessed our framework in two key scenarios: training models from scratch and explaining fully trained networks. In both cases, LAPs served dual roles: as external modules for post-hoc explanation and as plugged-in modules providing intrinsic interpretability. Experiments across two challenging datasets, one being medical, demonstrated that our framework consistently outperformed well-known white-box explainers in faithfully detecting decision-influencing regions. LAPs further enabled insightful per-case and layer-wise diagnosis of models, while the performance of other XAI methods significantly dropped when interpreting non-final convolutional layers. Moreover, incorporating LAPs during model training enhanced the model's interpretability, evident not only in LAP outputs but also in the quality of interpretations and explanations produced by other methods. To explore the potential effect of LAP as an internal module, we extended LAP as a pooling layer, and trained it alongside the model. The observations showed that LAP-extension did not result in deterioration in the performance, but slightly enhanced the average performance in most of the architectures across both datasets. Since the core of this study focuses on presenting a local interpretation method, we confined ourselves to exploring LAP's performance enhancement to the provided level, leaving further exploration for future studies.

Future work could investigate dynamic, per-sample ratio adjustment strategies to further improve both performance and interpretability beyond the results reported in this study. Such approaches could eliminate the need to set fixed, loss-related hyperparameters prior to training. Additionally, amid the current surge of interest in large language models (LLMs) and vision-language models (VLMs), model interpretability and explainability remains a significant challenge. This work could be further extended and systematically compared against existing XAI methods developed for these models.

# 7 Broader Impact Concerns

This work aims to improve the quality, soundness, and fidelity of interpretation maps, helping experts better understand the evidence influencing a model's prediction. Although validating a model's final prediction can be challenging, which is particularly crucial in critical applications such as medical imaging, XAI methods provide a way for the experts to inspect and validate whether the model relied on meaningful features.

However, XAI methods should be used strictly as supportive tools. A more plausible-looking or coherent explanation does not guarantee that the underlying prediction is correct, and explanation maps may themselves reflect spurious or misleading cues. These methods must therefore be applied with caution, particularly in high-stakes settings. We emphasize that the proposed approach is intended for human-in-the-loop workflows in which domain experts critically assess both predictions and their associated explanations, rather than as a mechanism for automated decision validation.

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

## A  Code Availability

The codes will be published as a Github repository upon acceptance. The codes are also uploaded with the main paper and supplementary material for the current submission. The codes, final trained models, training logs, and notebooks for running the codes are all available in this google drive . Notably, an anonymous account has been created for this end.

## B  Summary of Terms

Tab. 5 provides a summary of the terms used throughout the paper alongside their definitions and roles.

Table 5: Terms, their definitions and roles.

| Term | Meaning | Role |
|---|---|---|
| DC (Distinguishing Clue) | Minimal evidence region within input or intermediate feature maps that indicates a preference toward a particular subset of decisions over alternative options | Considered as probabilistic attribution maps of interpretation for the specified subset of decisions. |
| DC system | A set of DCs that collectively specify the regions within input or intermediate feature maps that promote each individual decision relative to all other possible decisions. | Providing interpretations in the form of probabilistic attribution maps for each individual decision. |
| LAP (Local Attention Perception) | A module that highlights the regions related to each DC for a target network layer | Produces importance maps for all DCs for the specified target layer. |
| LAP-Pool | Adopts LAP in the inner process of a pooling layer. | Leverages task-specific interpretations in the pooling operations. |
| LAP Supervision Loss | The loss used for training LAPs. | Training LAPs to automatically detect regions related to each DC without depending on expert annotations. |
| MinAR / MaxAR / IAR | Loss components of the LAP Supervision Loss that ensures coverage, precision, and exclusion respectively. | Training LAPs to precisely detect regions relevant to each DC in the samples that have the DC, while teaching it not to detect irrelevant regions in samples lacking the DC |

## C  The Difference Between LAP and Attention Mechanism

Unlike attention mechanisms, LAP does not apply a softmax function over the DC-wise importance maps for each sample. This design choice is motivated by the fact that the size of the affecting regions can vary substantially across samples. Applying a softmax over all DC-wise importance maps would assign lower per-pixel scores to large affecting regions compared to smaller ones, rendering the resulting scores incomparable in an absolute sense across different samples. This effect can be observed in Fig. 3. Indeed, all XAI methods except LAP have produced binarized attribution maps that highlight nearly the same number of pixels across cases, despite the fact that the expert-annotated affecting regions differ in size between samples.

From another perspective, LAP also does not apply a softmax function across different DCs at a single pixel location. A single pixel does not necessarily belong to only one DC; instead, a set of pixels collectively

forms a DC within an image. For example, as shown in Fig. 4, pixels belonging to the yellow spaceship have incorporated into the spaceship DC, while some of those same pixels, due to their proximity to building windows, have also incorporated into the taxi DC.

Instead, LAP applies a sigmoid function independently to each DC at each pixel, estimating the probability that a given pixel is incorporated into a particular DC. This formulation yields probabilistic outputs with an absolute meaning, making the scores directly comparable across different samples.

Another fundamental distinction between our LAP and standard attention mechanisms lies in how they treat individual elements. In conventional attention, multiple elements are typically aggregated into a single feature vector (e.g., through a weighted sum using attention weights). The loss is then either applied to this aggregated representation, or the feature map is incorporated into the architecture in a way that indirectly trains the attention weights. In contrast, LAP performs no such aggregation. Each element in each feature map is treated independently, and the proposed LAP-Supervision loss is applied directly to these per-element scores. This represents a fundamental departure from the aforementioned approaches.

## C.1 Formal Equations of LAP-Pool

We adopt the unified pooling framework of Gao et al. (Gao et al., 2019) to formalize the LAP-Pool procedure in Eq. (6). $O_{a',b'}^l$ represents the output corresponding to a pooling sliding kernel of size $W_K \times H_K$ positioned with its top-left corner at $(a,b)$, computed from the input feature map under the kernel $X_{a:a+H_K,b:b+W_K}^l$ and the weighting function $\mathscr{W}^l$. Here $\odot^*$ denotes element-wise multiplication with broadcasting, where the three-dimensional tensor $X_{a:a+H_K,b:b+W_K}^l$ is multiplied by the two-dimensional tensor $\mathscr{W}^l(X_{i:i+H_K,j:j+W_K})$.

$$\begin{cases} O_{a',b'}^l = \dfrac{\Sigma(\mathscr{W}^l(X_{a:a+H_K,b:b+W_K}^l) \odot^* X_{a:a+H_K,b:b+W_K}^l)}{\Sigma(\mathscr{W}^l(X_{a:a+H_K,b:b+W_K}^l))} \\ \mathscr{W}^l : \mathbb{R}^{F \times H_K \times W_K} \to R^{H_K \times W_K} \end{cases} \tag{6}$$

In this work, $\mathscr{W}^l$ is composed of two steps: pixel-wise scoring, $\mathscr{S}^l$, and local kernel-wise normalization, $\mathscr{N}^l$, according to Eq. (7). Pixel-wise importance scores ($Q_{i,j}^l$) are calculated by aggregating DC-wise importance probability maps ($I_{ij}^l$) using either a trainable or fixed function such as sum or max (denoted by $\mathscr{A}^l$). The aggregated scores within each pooling kernel region ($Q_{a:a+H_K,b:b+W_K}^l$) are then normalized locally to produce the final pixel-wise weights. Any suitable local normalization method may be adopted here. In our case, we use a Gaussian kernel centered at the pixel with the highest local score, allowing sensitivity to the most highlighted pixel to be modulated via a trainable parameter $\alpha$. When $\alpha = 0$, $\mathscr{N}(V) = 1 + \epsilon$, and as $\alpha \to \infty$, the coefficient of pixels other than $\max V$ approaches $\epsilon$, and LAP conveys only the features of the most highlighted local pixel. A small constant $\epsilon$ is added to avoid zero weights, ensuring gradient flow across all pixels and preventing division by zero during weighted averaging in (Eq. (6)).

$$\begin{cases} \mathscr{W}^l(X_{a:a+H_K,b:b+W_K}^l) = \mathscr{N}^l(Q_{a:a+H_K,b:b+W_K}^l) \quad ; \quad Q_{i,j}^l = \mathscr{A}^l(I_{i,j}^l) \\ \mathscr{N}^l(Q_{a:a+H_K,b:b+W_K}^l)_{i,j} = e^{-\alpha^2(\max(\{Q_{a:a+H_K,b:b+W_K}^l\}) - Q_{i,j}^l)^2} + \epsilon \\ \mathscr{N}^l : \mathbb{R}^{H_K \times W_K} \to \mathbb{R}^{H_K \times W_K} \quad ; \quad \mathscr{A}^l : \mathbb{R}^M \to \mathbb{R}^1 \end{cases} \tag{7}$$

## C.2 Integrating LAPs' scores to one unified attribution map

LAPs can be considered a sequence of information through the depth of the network. Shallower LAPs cannot capture enough information due to their low receptive field. Therefore, they are likely to make more mistakes. Some pixels may have been assumed to be important but were found unimportant in the deeper layers and vice versa. However, they produce more detailed maps because of the low receptive field and high resolution. We devised an algorithm to integrate interpretations from the final LAP layers iteratively to the initial layers. In this way, we can have both accuracy and resolution. The procedure's pseudo-code is presented in Algorithm 1, in which $\alpha$ is the decay factor to adjust the impact of shallower LAPs. Iteratively, the algorithm

modifies the current integrated map, $R_{l+1}$, with the current LAP attention map, $P_l$. Considering one pixel of $R_{l+1}$, $r_{l+1} \in \mathbb{R}$, and the set of its corresponding pixels in $P_l$, $p_l \in \mathbb{R}^{H_K \times W_K}$, if $r_{l+1}$ is active, i.e., greater than 0.5, at least one pixel in the corresponding zone must have been responsible. If any pixel of the $p_l$ is active, the credit only belongs to the active pixels. Otherwise, the current LAP does not comprehend the importance of this zone. Therefore, the credit belongs to all of them. Using this scheme, we prune the produced importance map from the last LAP, which is expected to be the most accurate, to the first. When choosing the topK pixels based on the ground truth bounding box size, the scores have been added without being clipped to keep the order of scores below 0.5 and have a proper selection.

---

**Algorithm 1** Pseudo-code for integrating the currently integrated map pixel on the position $(i, j)$ from the $L^{\text{th}}$ LAP to $l + 1^{\text{th}}$ LAP with $l^{\text{th}}$ LAP attention map in the corresponding kernel. This procedure is repeated for each pixel of each LAP layer, from $L$ to 1.

---

1: $\alpha \leftarrow 0.8$           ▷ the impact decay factor
2: $L \leftarrow$ The number of LAP layers
3: **procedure** INTEGRATEPIXEL$(R, P, l, i, j, H_K, W_K)$
4:     $p_l \leftarrow$ GETKERNEL$(P_l, i, j, H_K, W_K)$ ▷ $l^{\text{th}}$ LAP attention map for $(i, j)$'s corresponding kernel of size $H_K \times W_K$
5:     **if** $l = L$ **then**
6:        **return** $p_l$
7:     **end if**
8:     $r_{l+1} \leftarrow R_{l+1}[i, j]$          ▷ current integrated map pixel from $L^{\text{th}}$ LAP to $l + 1^{\text{th}}$ LAP
9:     $r_l \leftarrow r_{l+1}$ repeated to size $H_K \times W_K$          ▷ The result integrated map
10:     **if** $r_{l+1} \geq 0.5$ & $\max\{p_l\} \geq 0.5$ **then**
11:        **for** $i' : [0, H_K)$ **do**
12:           **for** $j' : [0, W_K)$ **do**
13:              $p \leftarrow p_l[i', j'] \times \alpha^{L-l}$          ▷ Apply the decay factor
14:              **if** $p_l[i', j'] \geq 0.5$ **then**
15:                 $r_l[i', j'] \leftarrow \max\{r_{l+1}, p\}$
16:              **else**
17:                 $r_l[i', j'] \leftarrow p$
18:              **end if**
19:           **end for**
20:        **end for**
21:     **end if**
22:     **return** $r_l$
23: **end procedure**

---

### C.3   A Note on Fidelity and Faithfulness in LAP-Based Interpretations

As discussed in the main text, LAP is a versatile framework for providing interpretations for the model. It can operate as an internal component within LAP-Pool modules, thereby directly incorporating interpretations into the model's decision-making process, or as a detached module that explains already trained networks without modifying their weights, intermediate representations, or decision mechanism.

The proposed attribution mechanism relies exclusively on intermediate feature maps extracted from different depths of the network rather than on the network weights themselves. In this setting, LAP learns the relationship between internal feature representations and potential model outcomes. This design primarily targets interpretation fidelity, as the explanations are derived from representations that are directly used by the model during inference. Although intermediate feature maps are not themselves the parameters governing the model's decision-making process, they are the direct outputs produced by that process up to a given depth of the network and constitute the only information available to subsequent layers. More formally, if the network is partitioned at section $\ell$, then given the intermediate representations at that section $(X^\ell)$, the model's final output becomes independent of both the original input $(X)$ and the parameters preceding that

layer ($\theta^{\ell-}$). Consequently, interpretations derived from these representations reflect how the model's internal processing has transformed the input information up to that point.

Therefore, instead of directly analyzing all parameters involved in the decision-making process, LAP-based interpretation traces behavioral faithfulness based on the outcome of intermediate calculations (intermediate feature maps) occurring during the real decision-making process of the model. In particular, feature maps computed near the final layers capture representations that already encode most of the decision-making process, making the output independent of earlier parameters and transformations. By analyzing how potential evidence regions for each output class (DCs) appear and persist across network depth, the method reveals which potential regions have been ignored through the model's internal reasoning between any two sections, and which ones have been kept as potentially related to the output throughout the depth. In practice, while the framework is fidelity-oriented, the resulting interpretations exhibit superior empirical faithfulness relative to baseline XAI methods, as measured by standard deletion and insertion metrics across multiple architectures.

### C.4   Pseudo-code for Training with LAP Modules

The general algorithm for training LAPs is presented in Algorithm 2. The algorithm performs a forward pass through the network $f_\theta$, during which both the final network prediction and the intermediate feature maps at the layers to which the LAPs are attached are computed and retained. The intermediate input of each LAP, $X^\ell$, is then passed to the corresponding LAP module to produce DC-wise importance probability maps ($I^\ell$). For each LAP and each DC, the loss terms described in Eq. (4) and Eq. (5) are subsequently computed. In the MinAR and MaxAR loss terms in Eq. (5), the reference DC-wise score map, $I^\ell_{\text{ref}_m}$, is used in identifying the Top-$K_{\text{MinAR}}$ and Bot-$K_{\text{MaxAR}}$ elements. Finally, the total training loss is obtained by combining the main task loss of the network with the LAP-Supervision loss terms.

When LAPs are integrated as LAP-Pools within the network, the forward pass automatically provides the required feature maps to the corresponding LAPs and computes the DC-wise importance probability maps. In scenarios where the backbone network is not intended to be trained or fine-tuned, the steps related to computing the main task loss and updating the backbone parameters are omitted from the algorithm.

### C.5   Element Selection in LAP Supervision Loss

In the LAP supervision loss presented in Eq. (4), all three loss terms rely on selecting candidate elements based on their importance probabilities of the related DC ($I^{l,m}$). While it is possible to use the importance probabilities from a single LAP layer to train that same layer, our experiments show that aggregating importance probabilities across multiple LAP layers for ranking leads to more stable and accurate results.

We also observed that selecting the Top-K and Bot-K element sets based solely on the probabilities assigned by the DC head can be sensitive to the model's initial parameters, particularly when rankings are computed independently for each LAP layer. In such cases, LAP layers with large receptive fields often become biased toward incorrect regions of the input. To mitigate this issue, we introduced an auxiliary module, referred to as the *discriminative scoring module*, which shares the same architecture as the original scoring module. This module is used to determine the Top-K and Bot-K elements sets, based on the *sum of outputs from both scoring modules*. It is trained using the first and third terms of the LAP supervision loss from Eq. (4), with $K_{\text{MinAR}} = K_{\text{IAR}} = n_e$. To avoid injecting misleading gradients into the main model, especially since many input regions may be shared across different DCs, we detached the input to the discriminative scoring module during training.

### C.6   Detailed Expert Knowledge Injection in LAP Supervision Loss

Human experts make their decisions based on specific features of the input. For example, jaggedness is one of the factors considered in classifying a tumor as malignant. Neural networks trained freely may or may not capture all the relevant reasons behind their decisions. They may become biased toward a dominant feature in the training dataset and thus lose generalization. LAP provides an easy way to inject expert knowledge into the network due to the probabilistic behavior of its scoring module. Experts can highlight the important

---

**Algorithm 2** Training with LAP Modules and LAP-Supervision Loss

---

**Require:** Backbone network $f_\theta$, LAP modules $\{\text{LAP}^\ell_\phi\}$, dataset $\mathcal{D} = \{(X_n, y_n)\}$, main loss $L_{\text{main}}$, per-LAP loss weights $\{\lambda^\ell\}$, epochs $T$

1: **for** epoch = 1 to $T$ **do**
2:      **for** each minibatch $B \subset \mathcal{D}$ **do**
3:          $\hat{y}, \{X^\ell\} = f_\theta(x) \triangleright$ Computing predictions and intermediate feature maps serving as inputs to the LAPs

4:          **for** each LAP module $\ell$ **do**
5:              $I^\ell \leftarrow \text{LAP}^\ell(X^\ell)$             $\triangleright$ *LAP outputs: DC-wise importance probability maps*
6:          **end for**
                         $\triangleright$ LAP-Supervision loss for each LAP
7:          **for** each LAP module $\ell$ **do**
8:              **for** each DC $\text{dc}_m$ **do**
9:                  $I^\ell_{\text{ref}_m} \leftarrow \text{build\_reference\_score\_map}(\{I^\ell_m\}, m, y)$     $\triangleright$ *Reference score map*
                            $\triangleright$ e.g., DC-wise sum: $I^\ell_{\text{ref}_m} = \sum_\ell I^\ell_m$
10:                  $L^{\text{MinAR}}_{\ell,m} \leftarrow \text{MinAR}(I^\ell, I_{\text{ref}}, m, y)$      $\triangleright$ *Only for samples that $\in \text{dc}^+_m$, i.e. $y_n \in Y^{\text{dc}^+_m}$*
11:                  $L^{\text{MaxAR}}_{\ell,m} \leftarrow \text{MaxAR}(I^\ell, I_{\text{ref}}, m, y)$      $\triangleright$ *Only for samples that $\in \text{dc}^+_m$, i.e. $y_n \in Y^{\text{dc}^+_m}$*
12:                  $L^{\text{IAR}}_{\ell,m} \leftarrow \text{IAR}(I^\ell, m, y)$            $\triangleright$ *Only for samples that $\in \text{dc}^-_m$, i.e. $y_n \in Y^{\text{dc}^-_m}$*
13:                  $L^{(\ell)}_{\text{LS}_{\ell,m}} \leftarrow L^{\text{MinAR}}_{\ell,m} + L^{\text{MaxAR}}_{\ell,m} + L^{\text{IAR}}_{\ell,m}$
14:              **end for**
15:          **end for**

16:          $L_{\text{total}} \leftarrow L_{\text{main}}(\hat{y}, y) + \sum_\ell \lambda_\ell \times (\sum_m L_{\text{LS}_{\ell,m}})$          $\triangleright$ *Total training loss*

17:          Update parameters: $\theta \leftarrow \theta - \eta\nabla_\theta L_{\text{total}}$    $\triangleright$ Or any other optimizer instead of SGD, like ADAM
18:          Update parameters: $\phi \leftarrow \phi - \eta\nabla_\theta L_{\text{total}}$    $\triangleright$ Or any other optimizer instead of SGD, like ADAM
19:      **end for**
20: **end for**
21: **return** trained model $f_\theta$, trained LAP modules $\{\text{LAP}^\ell_\phi\}$

---

input regions for their decision-making for each DC. The highlighted maps can be resized to match each LAP's input size and used as ground truth to train the DC scoring heads. The proposed LAP supervision loss enables the use of expert annotations instead of bootstrapped labels in the weak supervision process. For this end, in Eq. (5), in the $L^{\text{MinAR}}_{\text{LS}_{l,m}}$, and $L^{\text{MaxAR}}_{\text{LS}_{l,m}}$ loss terms, the exact positive and negative elements annotated by expert can be used instead of Top-$\text{K}_{\text{MinAR}}(I^{l,m})$ and Bot-$\text{K}_{\text{MaxAR}}(I^{l,m})$ respectively. The third term should be kept the same to keep the balance but in this scenario $K_{\text{IAR}}$ can also be set equal to the average size of DC annotated by expert in all DC-positive cases. The altered loss terms are calculated as follows. Let $\hat{A}^{l,m}$ denote the binary expert annotation map, resized to the input size of the $l$-th LAP for the $m$-th DC. We define $\text{Pos}(\hat{A}^{l,m})$ as the set of pixels labeled 1, $\text{Neg}(\hat{A}^{l,m})$ as the set of pixels labeled 0, $n_e$ as the total number of elements in $X^l$, and $n^+_m$ as the total number of DC-present samples for the $m$-th DC. Using these definitions, the corresponding loss terms are:

$$L_{\mathrm{LS}_{l,m}}^{\mathrm{MinAR}}(X^l, y, \hat{A}^{l,m}) = \frac{-2}{|\mathrm{Pos}(\hat{A}^{l,m})|} \times \sum_{i \in \mathrm{Pos}(\hat{A}^{l,m})} \ln(I_i^{l,m})$$

$$L_{\mathrm{LS}_{l,m}}^{\mathrm{MaxAR}}(X^l, y, \hat{A}^{l,m}) = \frac{-1}{|\mathrm{Neg}(\hat{A}^{l,m})|} \times \sum_{i \in \mathrm{Neg}(\hat{A}^{l,m})} \ln(1 - I_i^{l,m}) \tag{8}$$

$$L_{\mathrm{LS}_{l,m}}^{\mathrm{IAR}}(X^l, y) = \frac{-1}{K_{\mathrm{IAR}}} \times \sum_{i \in \mathrm{Top\text{-}K}_{\mathrm{IAR}}^{l,m}} \ln(1 - I_i^{l,m}) \quad ; \quad K_{\mathrm{IAR}} = \lceil \frac{\sum_{(X^l, y, \hat{A}^{l,m}):y \in Y^{\mathrm{dc}+}_m} |\mathrm{Pos}(\hat{A}^{l,m})|}{n_m^+} \times n_e^l \rceil$$

## C.7  Inter-LAP Concordance Loss Term for Inter-Layer Knowledge Sharing

Intuitively, if one LAP layer detects a DC in a part of the input, subsequent layers should detect it as well. To encourage this, we optionally apply the Jensen-Shannon divergence loss, defined in Eq. (9), on the importance probability maps of the $m$-th DC for two consecutive LAPs, $l$ and $l+1$, for a single sample $(X, y)$. Here, $\mathcal{JS}(I^{l,m}, R(I^{l+1,m}))$ denotes the Jensen-Shannon divergence loss for importance probability maps $I^{l,m}$, and the resized $I^{l+1,m}$ (resized via binary interpolation to match the dimensions of $I^{l,m}$). In this equation $n_e$ denotes the total number of elements in the largest map. This loss encourages consecutive LAP layers to reinforce each other's detection of clues and is applied to every pair of consecutive layers in our experiments.

$$\mathcal{JS}(I^{l,m}, R(I^{l+1,m})) = \frac{1}{2n_e} \sum_{(i)} \left[ (I_i^{l,m} - R(I_i^{l+1,m})) \ln \frac{I_i^{l,m}}{R(I_i^{l+1,m})} + (R(I_i^{l+1,m}) - I_i^{l,m}) \ln \frac{1 - I_i^{l,m}}{1 - R(I_i^{l+1,m})} \right] \tag{9}$$

## C.8  Experiments Setup

The main goal of this study is to propose a practical interpretation method that is generally applicable to all neural networks, including those that have already been trained. Notably, other published papers in the literature on general XAI methods have mostly assessed their proposed method on the whole version or a part of the ImageNet dataset using architectures like ResNet (Fel et al., 2023; Achtibat et al., 2023; Lee et al., 2021a; Ghorbani et al., 2019; Selvaraju et al., 2017). Following the literature, we conducted the experiments using two well-known and widely used architectures, ResNet and InceptionV3. While both architectures have high performance, they have different core ideas. ResNet is famous for its residual connections that prepare uninterrupted gradient paths to prevent gradient fading. Inception-V3 is known for its multi-resolution analysis, which applies kernels of different sizes to the feature map at each network level. These differences shed more light on the performance of different XAI methods.

ImageNet was adopted as a widely used dataset for the choice of dataset in assessing XAI methods. Besides ImageNet, as the medical domain proposes challenges that do not exist in object detection datasets, we also adopted the RSNA Pneumonia Detection dataset in a binary classification task to assess the general applicability of local XAI methods.

As mentioned in the main text, the proposed local XAI method has been compared with five widely used white-box local XAI methods, one inherently interpretable method, and one concept-based local explainer method. We have not compared the interpretation method with methods not applicable to all CNNs in general (e.g., the methods developed for the transformers), the ones not applicable to already trained models (e.g., self-interpretation methods that alter the architecture), the ones that have not provided an algorithm for a unified interpretation (e. g. LIP (Gao et al., 2019)), the global XAI methods that have not compared their results with any local method (e.g. (Ghorbani et al., 2019; Zhang et al., 2021)), and the methods that depend on other explainers we have compared our results with (e.g., HINT (Wang et al., 2022)). Also, the human-centered XAI methods are set aside as they need a human for the task.

### C.8.1 The choice of evaluation method for assessing XAI methods

Various evaluation metrics have been utilized in different studies to evaluate XAI methods. Each has advantages and disadvantages, making each more appropriate for specific domains. In this study, we have employed two distinct evaluation methods for our two datasets based on the nature of their domain.

- RSNA pneumonia detection:

  For the RSNA dataset, the input images are X-rays of the entire chest, and different parts of the image find different meanings based on their surroundings. In this sense, the domain differs from an object recognition dataset, where individuals can comprehend the object from its parts. Therefore, evaluation methods that involve removing parts of the image may influence the model's decision-making process due to producing out-of-distribution images. Methods like blurring may also result in artifacts that resemble pneumonia. Alternatively, the XAI methods are expected to capture the most distinguishing parts between images that belong to different categories. All pneumonia regions are, in fact, the distinguishing parts. We have used the "object localization" (Lee et al., 2021a) to evaluate the XAI methods. To this end, the Intersection over Union (IoU) metric was calculated between the attribution maps calculated by different XAI methods and the bounding boxes provided by experts for pneumonia in true positive cases.

- ImageNet:

  Unlike the RSNA dataset, in which the images of the two classes were only different in "pneumonia regions," the ImageNet dataset has 1000 classes, some of which are highly similar and only differ in small details, like different classes of dogs. Therefore, the XAI methods are not expected to highlight the whole object as a distinguishing decision reason. So, IoU cannot be a good representative of performance.

## D Experiment Discussion

### D.1 Training configurations

### D.1.1 A General Guide for Selecting Hyper-parameters of LAP Supervision Loss

In the training method described in Sec. 2.3, three hyper-parameters are involved: $\gamma^{\text{MinAR}}$, $\gamma^{\text{MaxAR}}$, and $\gamma^{\text{IAR}}$. There are two main approaches for setting these hyper-parameters:

1. **Using domain knowledge:** If the approximate range of sizes for Distinguishing Clues (DCs) is known, this information can guide the setting of $\gamma^{\text{MinAR}}$ and $\gamma^{\text{MaxAR}}$. It is recommended to set $\gamma^{\text{MinAR}}$ based on the median size of the DCs, and $\gamma^{\text{MaxAR}}$ based on the maximum expected size. For $\gamma^{\text{IAR}}$, it is preferable to set it equal to $\gamma^{\text{MinAR}}$ to ensure that the contribution of each element in the IAR loss term is balanced with that of the MinAR term.

2. **Empirical tuning:** Similar to typical hyperparameter tuning, one can define a candidate set of values, train LAPs using each configuration, and select the one that yields the highest faithfulness on validation data. Again, it is recommended to set $\gamma^{\text{IAR}} = \gamma^{\text{MinAR}}$ to maintain balance in the loss terms. For this purpose, a method similar to the one used for assessing faithfulness on the ImageNet dataset (Sec. 3.3) can be applied.

### D.1.2 RSNA

The RSNA pneumonia detection dataset contains chest X-ray images of 8851 healthy people and 6012 patients with lung pneumonia. We randomly selected 81% of the data for training, 9% for validation, and 10%

For weak supervision, we used cross-entropy loss on the classification head, LAP supervision loss, and inter-LAPs concordance loss with weights of 1, 10 per LAP, and 1 per LAP pair, respectively. The weights of

LAP supervision losses were set to 10 as we observed fast over-fitting of the main classifier compared to the LAP modules on the validation dataset.

We used $\gamma^{\mathrm{MinAR}} = 0.1, \gamma^{\mathrm{MaxAR}} = 0.5, \gamma^{\mathrm{IAR}} = 0.1$ as the hyper-parameter values of the LAP supervision loss. The first two were set based on the possible range of DC sizes, i.e., infection, in the positive samples. We chose the median and maximum of the infection bounding boxes as the mentioned bounds. The third was set to 0.1 to keep the balance between the element-wise loss factor of IAR and MinAR. For full supervision, we used cross-entropy loss on the classification head alongside cross-entropy loss on LAPs. We considered experts' bounding boxes as the ground truth with the same weights for LAP modules as in the weak supervision.

We trained the models for 200 epochs and stopped the runs if no progress was achieved on the validation data for 100 consecutive epochs. We selected the model related to the epoch with the best performance on validation data as the final model. We used balanced batches of size 64 (32 healthy and 32 pneumonia samples) and the ADAM optimizer with an initial learning rate of $10^{-4}$ and a decay coefficient of $10^{-6}$ in training. The models were trained using the Pytorch framework on a GEFORCE RTX 2080 Ti GPU.

### D.1.3 ImageNet

For the ImageNet dataset, 3 different architectures were adopted: ResNet50, InceptionV3, and EfficientNetB0. Since the objects in the ImageNet dataset have variable sizes and many are as large as half the image, only two LAPs were used in the architectures — one replacing the final adaptive pool and the other replacing the last non-adaptive pooling or strided convolution. For training LAPs, we incorporated our proposed weakly supervised loss — comprising $\gamma^{\mathrm{MinAR}}$ and $\gamma^{\mathrm{IAR}}$ both set to 0.1, and excluding MaxAR — weighted by a factor of 0.1 alongside the primary cross-entropy loss. The $\gamma^{\mathrm{MinAR}}$ hyper-parameter was selected from the set [0.01, 0.1, 0.4] based on validation data, and as described in Sec. D.1.1, $\gamma^{\mathrm{IAR}}$ was set equal to it. The models were trained using a learning rate of $10^{-4}$ and a weight decay coefficient of $10^{-6}$, a batch size of 1000, and the simple augmentation of resize and crop. Training was conducted in PyTorch on an Nvidia Quadro RTX 8000 GPU.The details of training for LAP-extended versions that provide intrinsic interpretability and detached LAPs (DLAP) that provide post-hoc explanations for the original models are expressed as follows:

- **DLAP-ResNet50:** Detached LAPs were trained for one epoch.

- **LAP-ResNet50:** One LAP-Pool replaced the strided convolution in Layer4, feeding the output of Layer3, and another LAP-Pool replaced the final adaptive-pool. The architecture from Layer 4 and all layers after that were tuned for 10 epochs.

- **DLAP-Inception3:** Detached LAPs were trained for 5 epochs.

- **LAP-InceptionV3:** For this architecture, a LAP-Pool replaced the final adaptive pool and the LAP in the middle layer was used in the detached mode over the outputs of Mixed7a layer as the original InceptionV3 model was heavily overtrained and tuning large parts of it resulted in degraded performance. The altered model was initialized with the original InceptionV3 weights, and the trained detached LAPs from DLAP-InceptionV3. The replaced pool and all layers after it were tuned for 3 epochs.

- **DLAP-EfficientNetB0:** Detached LAPs were trained for 5 epochs.

- **LAP-EfficientNetB0:** Same as LAP-InceptionV3, for this architecture, a LAP-Pool replaced the final adaptive pool and the LAP in the middle layer was used in the detached mode over the outputs of layer6.0 as the original EfficientNetB0 model was heavily overtrained and tuning large parts of it resulted in degraded performance. The altered model was initialized with the original EfficientNetB0 weights, and the trained detached LAPs from DLAP-EfficientNetB0. The replaced pool and all layers after it were tuned for 3 epochs.

## D.2 Further Results on RSNA Pneumonia Detection

### D.2.1 RSNA Pneumonia Detection Full Performance Evaluation Metrics

We evaluated the performance of the models using five metrics: accuracy, sensitivity, specificity, Balanced Accuracy (BA), and F1-score. Sensitivity and specificity are recalls of positive and negative classes that are widely used in medical domains. BA is their average, which provides a fair metric for imbalanced datasets. The results are presented in Tab. 6.It is observed that using LAP-supervision loss terms has enhanced sensitivity while, as a trade-off, degrading specificity. This suggests that the loss has encouraged the models to detect more evidence associated with the positive class, at the cost of increased false positives in some cases. Nevertheless, when considering fair single metrics for imbalanced datasets, such as F1-score and balanced accuracy, LAP-extension has resulted in an overall improvement in average performance.

Table 6: Performance metrics on RSNA Pneumonia Classification task. We evaluated accuracy, sensitivity, specificity, balanced accuracy, and F1-score for the vanilla ResNet18 (RN) and Inception3 (I3) models, BotCL-extended models (BCL-), LAP-extended models trained using the proposed weak supervision (WSL-), and LAP-extended models trained using expert annotations (BBL-). LAP-extended variants have achieved higher mean performances compared to the vanilla ones. All values are reported as mean ± standard deviation over five independently trained models initialized with different random seeds.

| Model | Accuracy | Sensitivity | Specificity | Balanced Accuracy | F1 |
|---|---|---|---|---|---|
| Org ResNet 18 | 94.5±0.24 | 93.63±0.95 | 95.42±0.67 | 94.52±0.25 | 94.58±0.27 |
| BCL-RN | 94.72±0.26 | 93.69±0.29 | **95.80**±0.31 | 94.74±0.26 | 94.79±0.26 |
| WSL-RN | 94.73±0.26 | **95.11**±0.73 | 94.34±1.11 | 94.72±0.27 | 94.88±0.22 |
| BBL-RN | **94.84**±0.19 | 94.12±1.44 | 95.6±1.56 | **94.86**±0.2 | **94.92**±0.19 |
| Inception V3 | 95.1±0.4 | 93.71±0.82 | **96.57**±0.33 | 95.14±0.39 | 95.15±0.41 |
| BCL-I3 | 95.15±0.27 | 94.29±0.54 | 96.05±0.85 | 95.17±0.28 | 95.22±0.25 |
| WSL-I3 | 95.22±0.18 | 93.97±0.42 | 96.55±0.27 | **95.25**±0.17 | 95.28±0.19 |
| BBL-I3 | **95.25**±0.21 | **94.33**±1.22 | 96.21±1.1 | 95.21±0.3 | **95.32**±0.24 |

### D.2.2 Different Choices of Binarization Approaches for Attribution Maps

Binarized attribution maps provide a common sense for the important regions in decision-making. Labeling a region as important or not is crucial in vital applications like medical diagnosis for validating the model's decision. For example, in breast cancer classification, regions highlighted as important will be analyzed in further medical diagnosis. Therefore, the region of extension and multi-focal importance detection are important in this field. Accordingly, in pneumonia detection, we focused on assessing the order of importance provided by attribution maps and binarized attribution maps.

Various binarization approaches have been adopted in the literature, from selecting a fixed portion, e.g., 15% of the top-scored pixels (Selvaraju et al., 2017), to adopting a threshold of mean + standard deviation ("$\mu + \sigma$") over the individual attribution maps (Lee et al., 2021a). We utilized three different approaches to assess attribution maps from different points of view. The results for the adoption of "$\mu + \sigma$" as threshold were presented in the main text (Tab. 2). Focusing on the other two approaches, in the first method, following the option of the fixed portion of highly scored pixels, rather than using a fixed percentage for all images, the number of pneumonia pixels known from the expert bounding boxes was used for each image (**Binarization by Top-Scored Selection**). We believe this results in a more fair comparison as the images in the dataset are highly diverse in the size of their pneumonia bounding boxes. The results are presented in Tab. 7. It is observed that LAP has achieved significantly higher performance than the other methods in all models. (p-value ≤ 0.001 using a paired t-test for original architectures and WS variants. BBs are excluded to provide a more fair results.)

Although binarization by Top-Scored Selection allows for a fair comparison of the order of scores by considering the size of the distinguishing area, the information is not known in test-time interpretation. Therefore, we also adopted a global thresholding method for binarization to assess the attribution maps (**Binarization**

**by Thresholding**). Notably, LAPs inherently provide a classification over the distinguishing pixels. We first normalized each importance map by its maximum value to find a global threshold for attribution maps calculated by other XAI methods. Then, we created a dataset from the normalized pixel-wise attribution maps over the validation samples. We assigned the positive label to all the pixels under the experts' annotated boxes and the negative to the others. We used RidgeClassifier of sklearn (Pedregosa et al., 2011) to classify the pixels. We also used balanced weighting to address the issue with the highly imbalanced dataset. We used the point with the prediction label equal to zero as the threshold for binarization. We applied this method for each trained model separately and used the resulting threshold to evaluate the attribution map calculated for the model. The results are presented in Tab. 8. Like the other methods, LAPs lead to significantly higher IoU scores than the other methods. (p-value $\leq 0.001$ using a paired t-test for original architectures and WS variants. BBs are excluded to provide a more fair results.)

Table 7: Assessment of binarized attribution maps using the "Binarization by TopK Selection" approach for the vanilla models of ResNet18 and Inception3, BotCL-extended models (BCL-), LAP-extended models trained by the proposed weak supervision (WSL-), and expert annotations (BBL-). While LAP-extension has enhanced the models' interpretability, LAP provides significantly better interpretations compared to the other XAI methods across all of the models. All values are reported as mean $\pm$ standard deviation over five independently trained models initialized with different random seeds.

| Model | IoU Between Expert Annotations and Binarized Attribution Maps (TopK Selection) | | | | | | | |
|---|---|---|---|---|---|---|---|---|
|  | DL | GGC | GC | GBP | RC | CR | BotCL | LAP |
| ResNet18 | 19.69±0.98 | 19.56±1.35 | 27.17±1.95 | 16.74±1.24 | 9.86±1.37 | 27.77±2.07 | NA | **33.70**±3.48 |
| BCL-RN | 21.77±1.08 | 17.12±2.58 | 20.76±2.60 | 20.84±1.17 | 25.02±5.02 | 24.42±4.35 | 22.70±3.71 | **26.72**±2.66 |
| WSL-RN | 22.40±0.61 | 21.80±0.84 | 18.37±1.95 | 21.54±0.66 | 41.35±1.22 | 39.42±3.41 | NA | **41.96**±1.43 |
| BBL-RN | 22.60±0.78 | 24.08±0.86 | 24.70±4.83 | 23.03±1.27 | 31.02±4.66 | 43.74±4.66 | NA | **55.95**±1.46 |
| Inception V3 | 23.42±0.33 | 24.09±1.60 | 32.69±1.24 | 22.30±1.23 | 33.96±1.42 | 32.14±1.83 | NA | **43.34**±1.48 |
| BCL-I3 | 22.69±0.85 | 18.32±8.33 | 8.13±5.81 | 23.56±1.32 | 13.39±4.42 | 5.88±6.27 | 5.17±2.50 | **26.93**±9.42 |
| WSL-I3 | 22.90±0.17 | 14.55±2.91 | 8.80±2.18 | 14.71±3.24 | **41.30**±0.70 | 40.81±0.88 | NA | 39.06±0.84 |
| BBL-I3 | 24.12±0.50 | 19.11±3.40 | 25.35±3.88 | 18.03±3.22 | 45.88±0.66 | 38.85±5.38 | NA | **55.72**±0.48 |

Table 8: Assessment of binarized attribution maps using the "Binarization by Thresholding" approach for the vanilla models of ResNet18 and Inception3, BotCL-extended models (BCL-), LAP-extended models trained by the proposed weak supervision (WSL-), and expert annotations (BBL-). While LAP-extension has enhanced the models' interpretability, LAP provides significantly better interpretations compared to the other XAI methods across all of the models. All values are reported as mean $\pm$ standard deviation over five independently trained models initialized with different random seeds.

| Model | IoU Between Expert Annotations and Binarized Attribution Maps (Learned Threshold) | | | | | | | |
|---|---|---|---|---|---|---|---|---|
|  | DL | GGC | GC | GBP | RC | CR | BotCL | LAP |
| ResNet18 | 16.37±0.77 | 16.28±1.05 | 21.62±1.94 | 13.89±0.90 | 11.75±0.64 | 24.55±1.38 | NA | **31.37**±2.79 |
| BCL-RN | 17.89±1.31 | 13.65±1.90 | 18.33±2.34 | 16.36±1.12 | 19.91±3.14 | 21.41±3.68 | 20.28±2.92 | **25.01**±2.48 |
| WSL-RN | 18.91±0.56 | 18.45±0.82 | 17.40±1.45 | 18.44±0.71 | 34.91±0.97 | 32.82±3.55 | NA | **36.28**±1.19 |
| BBL-RN | 19.02±0.80 | 20.57±0.65 | 18.54±1.05 | 19.76±1.03 | 24.93±2.44 | 31.91±8.25 | NA | **43.26**±1.51 |
| Inception V3 | 19.26±0.40 | 16.99±1.38 | 25.49±1.46 | 16.60±1.23 | 27.05±1.49 | 25.55±0.69 | NA | **29.52**±0.93 |
| BCL-I3 | 18.47±0.84 | 12.69±5.50 | 8.79±5.19 | 16.70±1.30 | 16.05±2.40 | 4.10±5.41 | 3.06±1.46 | **24.40**±2.05 |
| WSL-I3 | 19.32±0.26 | 12.47±1.82 | 8.65±1.08 | 12.84±2.22 | 32.60±0.98 | 33.60±1.57 | NA | **35.75**±0.62 |
| BBL-I3 | 20.49±0.55 | 15.30±2.29 | 17.03±4.08 | 14.93±2.28 | 35.05±2.37 | 26.40±5.36 | NA | **41.46**±1.87 |

### D.2.3 SRG scores for assessing faithfulness of XAI methods on RSNA dataset

To further assess the faithfulness of XAI methods to the underlying model on the RSNA dataset, we computed SRG scores (Bluecher et al., 2024) for all XAI methods across all architectures. Although perturbation-based evaluations such as pixel flipping can introduce severe out-of-distribution artifacts in medical images, the original study demonstrated that SRG-based rankings are comparatively robust to such distribution shifts and remain consistent across different evaluation configurations. Because the largest expert-annotated bounding boxes cover at most half of the image area, using lower retention ratios would result in heavily corrupted images that are not meaningful in a medical context. Therefore, we computed the SRG score only over

retention ratios between 0.5 and 1 (specifically: 0.5, 0.6, 0.7, 0.8, 0.9, 0.925, 0.95, 0.975, 0.99) and quantified it as the area under this segment of the curve. Consequently, the maximum attainable SRG score in our setting is 0.5. The results are reported in Tab. 9.

Interestingly, these findings are fully consistent with the IoU-based evaluation presented in Tab. 2. While IoU measures the spatial agreement between binarized attribution maps and expert-provided bounding boxes, regions that the model may not necessarily rely on, SRG score directly evaluates the faithfulness of explanation methods to the model's prediction. Across all architectures, LAP achieved the highest SRG scores. This effect is particularly notable for BotCL-extended architectures. Although BotCL is designed to be inherently interpretable, and detached LAPs are used as post-hoc explanations for BotCL-extended models, LAP nevertheless produces substantially higher SRG scores than BotCL's native interpretations (0.24 and 0.24 vs. 0.11 and 0.01 for BCL-ResNet18 and BCL-InceptionV3, respectively). Furthermore, adopting the proposed training scheme for LAP-based explanations improves SRG scores not only for LAP but also for other explanation methods, such as Relevance-CAM and CRAFT. This suggests that the proposed training strategy enhances the intrinsic interpretability of the trained models.

Table 9: SRG scores computed from attribution maps calculated by different XAI methods for vanilla models of ResNet18 and Inception3, BotCL-extended models (BCL-), LAP-extended models trained by the proposed weak supervision (WSL-), and expert annotations (BBL-), for the cases predicted as positive in the RSNA test set. Higher scores indicate greater faithfulness of the XAI method to the underlying decision-making process of the model. LAP provides significantly better interpretations compared to the other XAI methods across all of the models. All values are reported as mean $\pm$ standard deviation over five independently trained models initialized with different random seeds.

| Model | DL | GGC | GC | GBP | RC | CR | BotCL | LAP |
|---|---|---|---|---|---|---|---|---|
| ResNet18 | 0.01±0.00 | 0.01±0.00 | 0.20±0.10 | 0.01±0.00 | 0.01±0.00 | 0.22±0.09 | - | **0.33**±0.07 |
| BCL-RN | 0.01±0.02 | 0.01±0.00 | 0.11±0.11 | 0.01±0.00 | 0.06±0.08 | 0.18±0.15 | 0.11±0.07 | **0.24**±0.12 |
| WSL-RN | 0.01±0.00 | 0.01±0.00 | 0.18±0.07 | 0.01±0.00 | 0.39±0.03 | 0.36±0.04 | - | **0.42**±0.02 |
| BBL-RN | 0.01±0.01 | 0.01±0.02 | 0.19±0.11 | 0.01±0.01 | 0.31±0.05 | 0.32±0.10 | - | **0.38**±0.05 |
| Inception V3 | 0.01±0.00 | 0.01±0.02 | 0.09±0.08 | 0.01±0.02 | 0.10±0.09 | 0.09±0.09 | - | **0.23**±0.11 |
| BCL-I3 | 0.01±0.00 | 0.01±0.00 | 0.08±0.10 | 0.01±0.00 | 0.02±0.01 | 0.02±0.03 | 0.01±0.00 | **0.24**±0.20 |
| WSL-I3 | 0.01±0.00 | 0.01±0.00 | 0.06±0.03 | 0.01±0.00 | 0.40±0.01 | 0.40±0.01 | - | **0.47**±0.01 |
| BBL-I3 | 0.01±0.00 | 0.01±0.00 | 0.27±0.04 | 0.01±0.00 | **0.40**±0.02 | 0.37±0.04 | - | **0.40**±0.02 |

### D.2.4 Other metrics for assessing XAI methods on RSNA

To provide a more comprehensive assessment of attribution maps for the RSNA dataset calculated by different XAI methods, we have adopted two additional metrics alongside the standard IoU calculation with expert bounding boxes. Given that bounding boxes are inherently rectangular and may not perfectly align with the actual infected area, more flexible evaluation metrics may help in shedding light on the performance of XAI methods. In this regard, the sensitivity of bounding box capturing and truncated IoU were adopted as complementary metrics. For sensitivity, we have considered a bounding box 'captured' if at least half of it is covered by the binarized attribution map, reporting the percentage of captured boxes relative to the total. The truncated IoU only includes half of each bounding box containing the highest-scored pixels detected by each method in calculating IoU, ensuring that a hypothetical perfect method would achieve an IoU score greater than or equal to any imperfect method. Both metrics utilize either binarized maps generated using the dynamic threshold of $\mu + \sigma$ or the automatically learned threshold closest to the ground truth map. Results presented in Tab. 10 and Tab. 11 demonstrate that LAP outperforms all other methods across both metrics, showcasing its superiority from multiple perspectives.

### D.2.5 Ablation Study on the Hyper-parameters of the LAP Supervision Loss

We conducted another ablation study on the RSNA dataset using the ResNet-18 network to assess the effect of different values for the hyperparameters in the LAP-supervision loss function (Eq. (4)). The results are presented in Tab. 12. In eight settings, the network was trained with different values of $\gamma^{\mathrm{MinAR}}$, $\gamma^{\mathrm{MaxAR}}$, and $\gamma^{\mathrm{IAR}}$. The settings were named as triplets of $[\gamma^{\mathrm{MinAR}}][\gamma^{\mathrm{MaxAR}}][\gamma^{\mathrm{IAR}}]$, where $O$ stands for the default

Table 10: Assessment of the sensitivity of the XAI methods regarding capturing expert bounding boxes for the vanilla models of ResNet18 and Inception3, BotCL-extended models (BCL-), LAP-extended models trained by the proposed weak supervision (WSL-), and expert annotations (BBL-). While LAP-extension has enhanced the models' interpretability, LAP provides significantly better interpretations compared to the other XAI methods across all of the models. All values are reported as mean ± standard deviation over five independently trained models initialized with different random seeds.

| Model | Sensitivity of Capturing Expert Bounding Boxes | | | | | | | |
|---|---|---|---|---|---|---|---|---|
| | DL | GGC | GC | GBP | RC | CR | BotCL | LAP |
| ResNet18 | 0.06±0.10 | 0.05±0.06 | 46.75±4.17 | 0.00±0.00 | 9.12±2.68 | 48.43±4.49 | NA | **57.82**±6.12 |
| BCL-RN | 0.12±0.14 | 0.76±0.42 | 32.33±3.96 | 0.29±0.21 | 36.10±8.95 | 31.27±9.77 | 29.67±7.51 | **41.66**±6.98 |
| WSL-RN | 0.01±0.03 | 0.09±0.09 | 24.16±4.10 | 0.13±0.17 | 76.56±4.17 | 75.76±4.20 | NA | **79.43**±7.89 |
| BBL-RN | 0.03±0.03 | 0.16±0.16 | 36.16±12.85 | 0.04±0.05 | 52.06±8.03 | 71.81±4.79 | NA | **74.36**±2.32 |
| Inception V3 | 0.00±0.00 | 0.06±0.13 | 58.37±2.71 | 0.01±0.03 | 60.30±3.78 | 55.43±1.67 | NA | **79.40**±3.35 |
| BCL-I3 | 0.00±0.00 | 0.01±0.03 | 4.88±7.13 | 0.03±0.05 | 15.07±12.50 | 8.11±12.81 | 0.85±0.94 | **47.74**±16.90 |
| WSL-I3 | 0.00±0.00 | 0.00±0.00 | 13.67±4.17 | 0.00±0.00 | 72.49±2.63 | 69.48±2.51 | NA | **78.57**±3.33 |
| BBL-I3 | 0.00±0.00 | 0.00±0.00 | 43.82±10.11 | 0.00±0.00 | 70.77±4.63 | 59.31±6.37 | NA | **73.66**±2.23 |

Table 11: Assessment of the XAI methods by calculating truncated IoU for the vanilla models of ResNet18 and Inception3, BotCL-extended models (BCL-), LAP-extended models trained by the proposed weak supervision (WSL-), and expert annotations (BBL-). While LAP-extension has enhanced the models' interpretability, LAP provides significantly better interpretations compared to the other XAI methods across all of the models. All values are reported as mean ± standard deviation over five independently trained models initialized with different random seeds.

| Model | Truncated IoU Between Expert Annotations and Binarized Attribution Maps (Best Thresholding) | | | | | | | |
|---|---|---|---|---|---|---|---|---|
| | DL | GGC | GC | GBP | RC | CR | BotCL | LAP |
| ResNet18 | 14.20±0.89 | 10.86±1.54 | 24.97±1.98 | 8.70±1.36 | 7.34±1.39 | 26.60±1.74 | NA | **35.04**±4.02 |
| BCL-RN | 15.32±1.53 | 10.08±1.61 | 19.45±2.38 | 11.53±1.09 | 22.18±5.29 | 21.34±5.09 | 19.94±3.90 | **26.38**±3.32 |
| WSL-RN | 16.55±0.68 | 13.95±0.82 | 15.83±2.02 | 14.10±0.75 | 41.19±1.57 | 39.32±3.18 | NA | **41.59**±1.63 |
| BBL-RN | 17.05±0.89 | 16.75±0.79 | 21.40±5.32 | 15.63±1.14 | 29.18±4.95 | 41.04±4.98 | NA | **56.70**±1.62 |
| Inception V3 | 16.18±0.49 | 13.66±1.76 | 29.83±1.60 | 12.21±1.67 | 31.96±1.69 | 30.16±1.24 | NA | **40.55**±1.59 |
| BCL-I3 | 15.15±0.87 | 9.72±4.50 | 3.79±3.57 | 13.15±1.56 | 9.57±4.85 | 4.09±5.91 | 1.90±1.16 | **26.13**±8.16 |
| WSL-I3 | 16.32±0.36 | 7.82±1.48 | 8.35±2.38 | 8.40±1.99 | 39.39±1.29 | 38.95±1.64 | NA | **40.94**±0.90 |
| BBL-I3 | 18.08±0.55 | 11.41±2.64 | 23.82±4.49 | 10.48±2.57 | 43.09±1.80 | 36.51±3.85 | NA | **56.76**±0.55 |

value, $L$ for a much lower value, and $H$ for a much higher value. The values of the hyperparameters are presented in Tab. 13. It was observed that the proposed setting, based on general domain knowledge according to Sec. D.1.2, outperformed the others in terms of both task-specific performance metrics and interpretation-based metrics. Compared to the classification performance of the base ResNet-18 architecture presented in Tab. 1, all settings achieved a higher average F1-score. Among the settings, three had lower F1-scores compared to the others: LOO, OHO, and HHH. It appears that using a much higher value for $\gamma^{\mathrm{MinAR}}$ has the largest negative effect on task-specific performance. Additionally, using a much lower value for $\gamma^{\mathrm{MinAR}}$ is also detrimental, as observed in the Naive weak-supervision V1 setting in Tab. 3.

Focusing on the IoU of binarized attribution maps, four settings exhibited drastically lower IoU after thresholding compared to the others: LOO, LLL, HOO, and HHH. It was observed that interpretability is primarily sensitive to a single hyperparameter: $\gamma^{\mathrm{MinAR}}$. Using either a much higher or much lower value for $\gamma^{\mathrm{MinAR}}$ results in a noticeable decrease in the IoU of the binarized attribution maps when self-thresholding is applied. This occurs because the chosen value represents the expected size of the distinguishing part, which allows the network to explore an appropriate portion of the image. A much higher value forces the network to select irrelevant pixels as distinguishing parts, deteriorating the gradient flow, while a much lower value prevents the network from exploring all distinguishing parts, causing it to focus only on highly distinctive pixels and reducing generalization. This experiment demonstrates that hyperparameters can be selected using the method described in Sec. D.1.2, without the need to try numerous values. Notably, all configurations except the ones mentioned above achieve substantially higher IoU compared to other XAI methods (Tab. 2).

Table 12: Task-specific performance metrics and IoUs of infection bounding boxes of RSNA and binarized attribution maps of the LAP ResNet 18 network trained with different configurations using two binarization approaches, thresholding and TopK selection by ground truth. Configurations are presented as triplets of $[\gamma^{\mathrm{MinAR}}][\gamma^{\mathrm{MaxAR}}][\gamma^{\mathrm{IAR}}]$, in which $O$ stands for the default value, $L$ for a lower value, and $H$ for a higher one. The selected configuration based on the general domain-specific knowledge has resulted in both high task-specific performance metrics and interpretation metrics. All values are reported as mean $\pm$ standard deviation over five independently trained models initialized with different random seeds.

| Training | Performance Metrics | | | | | IoU | |
|---|---|---|---|---|---|---|---|
| Configuration | Accuracy | Sensitivity | Specificity | B Acuuracy | f1 | Thresholding | TopK |
| Proposed | 94.73±0.26 | **95.11**±0.73 | 94.34±1.11 | 94.72±0.27 | 94.88±0.22 | 36.28±1.19 | 41.96±1.43 |
| LOO | 94.46±0.21 | 94.68±1.01 | 94.24±0.83 | 94.46±0.2 | 94.61±0.23 | 25.82±0.82 | 39.49±1.4 |
| OLO | 94.73±0.27 | 93.61±0.42 | **95.91**±0.76 | 94.76±0.29 | 94.8±0.26 | 36.08±0.65 | 42.19±1.23 |
| OOL | **94.86**±0.31 | 94.1±0.52 | 95.66±0.32 | **94.88**±0.3 | 94.94±0.31 | 36.16±2.99 | 43.15±3.48 |
| LLL | 94.75±0.29 | 94.25±0.79 | 95.28±0.81 | 94.76±0.29 | 94.85±0.29 | 24.32±2.47 | 42.15±0.78 |
| HOO | 94.73±0.15 | 94.44±0.32 | 95.03±0.55 | 94.74±0.16 | 94.84±0.13 | 26.05±0.5 | 39.21±3.32 |
| OHO | 94.57±0.26 | 94.61±0.62 | 94.51±0.7 | 94.56±0.26 | 94.69±0.25 | **36.64**±1.67 | **43.49**±2.25 |
| OOH | 94.85±0.27 | 94.51±0.32 | 95.21±0.3 | 94.86±0.27 | **94.95**±0.26 | 35.63±3.35 | 41.38±3.96 |
| HHH | 94.49±0.31 | 93.93±0.75 | 95.08±1.2 | 94.5±0.33 | 94.59±0.27 | 20.58±1.13 | 34.42±0.91 |

Table 13: The values used for $\gamma^{\mathrm{MinAR}}$, $\gamma^{\mathrm{MaxAR}}$, and $\gamma^{\mathrm{IAR}}$ hyper-parameters in the ablation study of RSNA.

| Type | O | L | H |
|---|---|---|---|
| $\gamma^{\mathrm{MinAR}}$ | 0.1 | 0.01 | 0.5 |
| $\gamma^{\mathrm{MaxAR}}$ | 0.5 | 0.1 | 0.9 |
| $\gamma^{\mathrm{IAR}}$ | 0.1 | 0.01 | 0.5 |

### D.2.6 Assessing the Effect of Aggregation in Interpretation

As mentioned in the main text, LAPs can be used separately to present a sectional interpretation for the network. The deeper the LAP, the closer and more representative to the final decider, but with a lower resolution that makes the exact locating of the distinguishing DCs difficult. As discussed in Sec. 2.2 and Sec. C.2, we adopted the hierarchical relations of LAPs to provide an aggregated attribution map that has benefited from the high resolutions of the shallower LAPs to enhance the interpretations of the deeper LAP. To assess the sectional interpretations provided by different LAPs, the IoU between the binarized attribution maps and the ground truth expert bounding boxes are calculated according to Tab. 14. The metric was also calculated for the aggregations of LAPs (Aggregated), which is the metric used all over the paper. Notably, the metric achieved by two LAPs in two different sections might represent the ability of the network to detect the exact location of infected zones, i.e. the distinguishing regions, in different depths. Therefore, we cannot compare the metric achieved by an aggregation with an LAP different from the aggregation's last layer. The metrics reported in the table indicate the ultimate suffering of the deepest LAP, i.e. LAP4, from the low resolution. It can be observed that using the aggregation has resulted in a great enhancement for LAP4 in all of the architectures. Additionally, in the proposed training scheme, with no expert supervision included, using the aggregation has resulted in the best DC locating compared to the isolated LAPs.

### D.3 Further Results on ImageNet Object Classification

### D.3.1 Comparing faithfulness of the XAI methods on ImageNet with respect to the predicted label

We compared the faithfulness of different XAI methods on each model by modifying the input images based on the attribution maps. Faithfulness was evaluated by keeping $k\%$ pixels with the highest scores provided by the XAI method, zeroing out the rest of the image, and then passing the modified image through the network for re-evaluation. The results are presented in Fig. 6 for percentages of 10%, 30%, 50%, 70%, and 90% across

Table 14: IoU between the binarized attribution maps based on thresholding for LAPs in different sections of the original architectures, LAP-extended architectures trained with the proposed weakly supervised method (WSL), and LAP-extended architectures trained with the infection bounding boxes annotated by experts (BBL). The results for the aggregated attribution maps are also presented as Aggregated. Comparing Aggregated, which is the one used in all comparisons in this paper, with LAP4, it can be observed that aggregation has enhanced the IoU scores by adopting the resolution of the previous layers. All values are reported as mean ± standard deviation over five independently trained models initialized with different random seeds.

| Model | LAP1 | LAP2 | LAP3 | LAP4 | Aggregated |
|---|---|---|---|---|---|
| ResNet 18 | 24.63±2.60 | 29.43±2.70 | **32.18**±1.64 | 25.22±0.36 | 31.37±2.79 |
| Inception V3 | 26.56±0.49 | **36.16**±1.10 | 25.40±0.65 | 24.74±0.69 | 29.52±0.93 |
| WSL ResNet | 34.92±2.04 | 36.14±0.98 | 33.37±1.36 | 25.47±1.69 | **36.28**±1.19 |
| WSL Inception | 29.30±0.90 | 35.64±0.62 | 33.42±0.46 | 25.82±0.34 | **35.75**±0.62 |
| BBL ResNet | 40.01±1.91 | **46.11**±1.07 | 45.44±1.14 | 39.96±0.46 | 43.26±1.51 |
| BBL Inception | 28.52±1.58 | **47.64**±1.14 | 44.13±1.84 | 40.65±0.98 | 41.46±1.87 |

six model variants: LAP-Extended and Original versions of ResNet50, InceptionV3, and EfficientNetB0 [4]. In this experiment, the attribution maps were calculated based on the predicted class, and the results were also evaluated with respect to the original predictions. A faithful XAI method would capture the influencing regions for decision-making in the model, thereby keeping the original decision for the modified image. We assessed the faithfulness of attribution maps calculated for two different target layers: the last convolutional layer before the Adaptive Pooling—commonly used by CAM-based methods—and an earlier convolutional layer that outputs a higher-resolution feature map than the final one (the layer before the last pooling). Our method demonstrates the highest faithfulness compared to other XAI techniques across both layers and all evaluated models, confirming it as the most faithful overall. While the competing method CradCam has slightly outperformed LAP in the case of EfficientNet with the target being the last convolutional layer, the differences are minimal. Notably, LAP provides significantly better attribution maps for the intermediate layer, highlighting the limitations of other methods in interpreting deeper layers, which is crucial for thorough model diagnosis.

---

[4]As CRAFT requires days to learn post-hoc explanations, it has only been compared across three architectures: LAP-ResNet50, Original ResNet-50, and LAP-EfficientNet-B0.

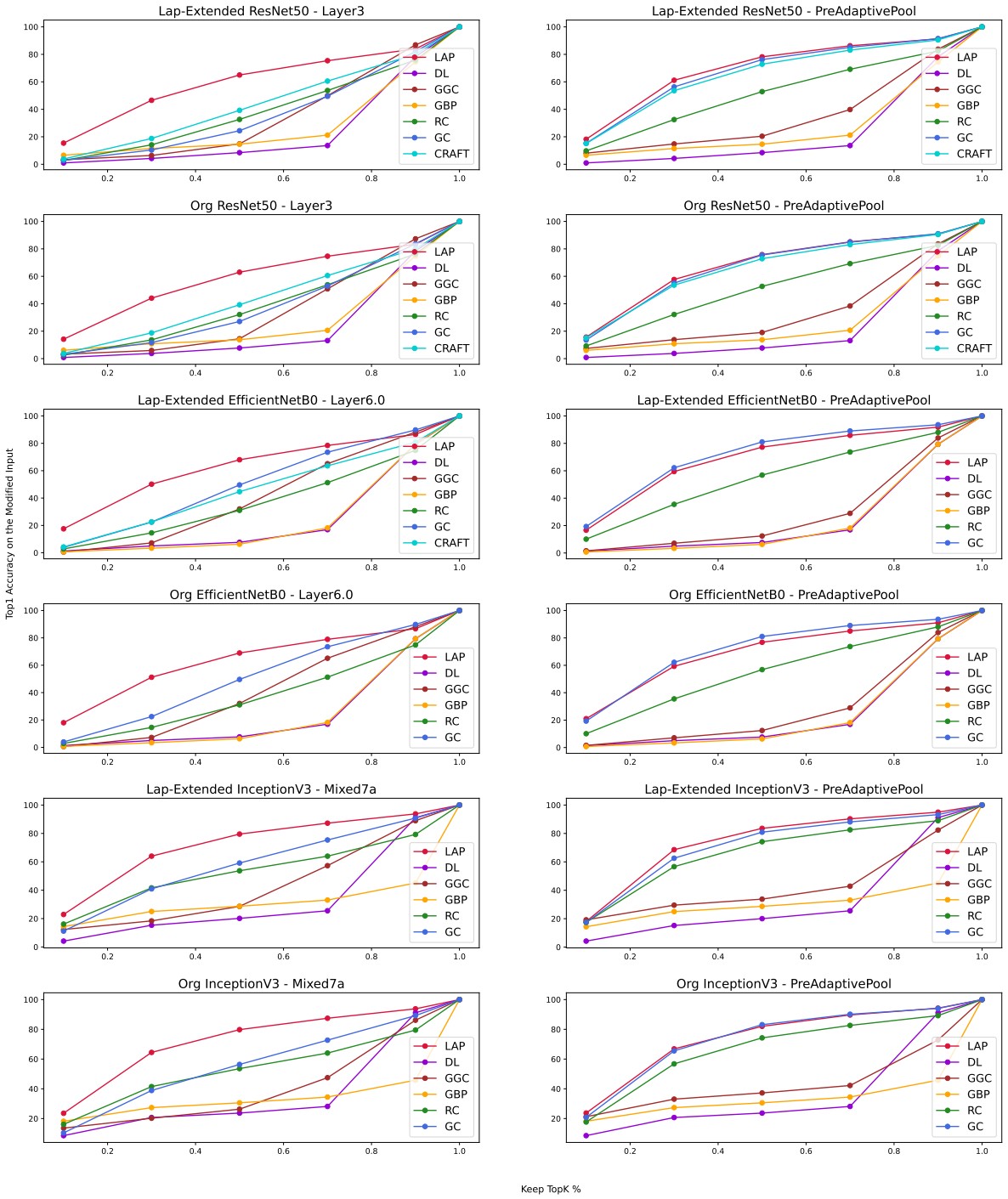

Figure 6: Assessing the faithfulness of different XAI methods on the ImageNet dataset for various models over the outputs of two different layers. Faithfulness is evaluated by keeping a ratio of pixels ranked by the XAI method (K%), zeroing out the rest of the image, and evaluating the prediction on the modified image with respect to the original prediction. LAP generally outperforms other methods, especially for middle-layer-based attribution maps; only in two cases, both with the target layer being the last convolutional layer, does the competing method CradCam slightly outperform LAP, while in the remaining ten cases LAP shows superior faithfulness.

### D.3.2 Comparing faithfulness of the XAI methods on ImageNet with respect to the ground truth

We compared the faithfulness of different XAI methods on the LAP-Extended ResNet 50 network (with layer3 as target of the interpretation) on the ImageNet dataset by modifying the input images based on the attribution maps calculated by different XAI methods. Faithfulness was evaluated by keeping $k\%$ of pixels with the highest scores provided by the XAI method and zeroing out the rest of the parts of the image, then passing the modified image as input through the network for re-evaluation. The results are presented in Fig. 7 for percentages of 10%, 30%, 50%, 70% and 90%. In this experiment, the attribution maps were calculated based on the ground truth class, and the results were also evaluated with respect to the ground truth. A faithful XAI method would capture the effective regions for decision-making in the model, keeping the former decision for the modified image. Our method represents the highest faithfulness than other XAI methods in Top1 and Top5 accuracies, introducing itself as the most confident.

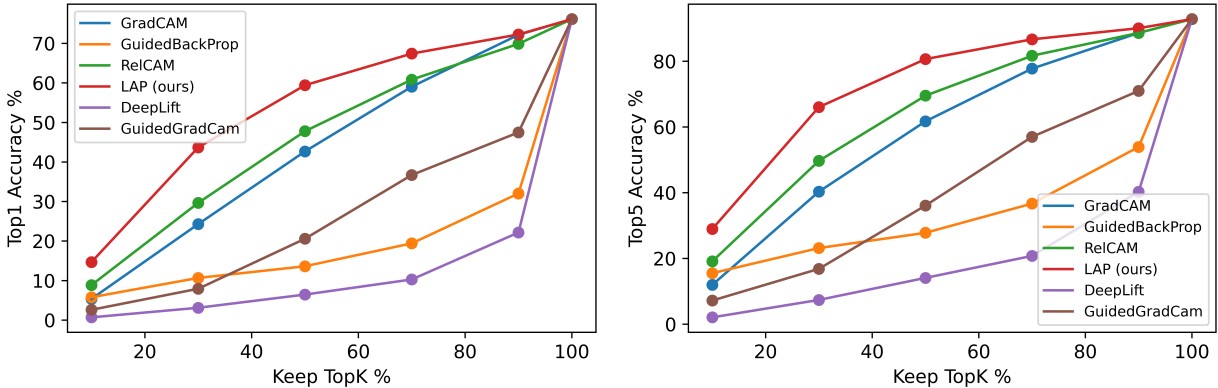

Figure 7: Assessing the faithfulness of different XAI methods on the ImageNet dataset for LAP-Extended ResNet 50 and layer 3 as the target. Faithfulness is evaluated by keeping a ratio of pixels with the highest scores provided by the XAI method (K%) for the ground truth class, then zeroing out the rest parts of the image, and evaluating the prediction based on the modified image with respect to the prediction of the original image. Our method in Top1 and Top5 accuracies exceeds the other methods.

## D.4 Assessing Execution Times

In terms of assessing computational efficiency, the required time for both phases, "training time" and "inference time," has been calculated for both datasets. In this section, inference time refers to the time needed to run the model to produce outputs and interpret those outputs, which must be practical for real-world applications. Conversely, training time should be manageable to make the method affordable for users. The following subsections include the details.

### D.4.1 Training Time

Tab. 15 represents the time required to train different versions of each model for one epoch on an RTX Quadro 8000 GPU. Comparing the times required to train original architectures with the LAP-extended versions, and considering the fact that LAPs only need a few epochs to be trained as explained in Sec. D.1.3. The training time for LAP is both reasonable and affordable. In the case of CRAFT, the required time is excessive and unaffordable. BotCL, on the other hand, requires training the architecture from scratch and therefore needs more than 100 epochs to be trained, making it as unaffordable as CRAFT for large datasets like ImageNet at small research centers and small companies.

### D.4.2 Inference Time

In order to examine whether extending the networks with LAP poses lingering execution time in the test phase, we calculated the time required for running the model and interpreting the results for different architectures

Table 15: Training times required for different architectures over 1 epoch for the original architecture, LAP-extended version trained with the proposed weakly supervised approach as a plugged-in module (WSL-), LAPs being tuned as post-hoc explainers (DLAP-), and BotCL-expanded version. In the case of CRAFT, the total required time has been calculated as the algorithm does not work by epochs. In some cases, the extended version of the model takes less time to be trained as portions of the original network have been frozen. The results indicate the feasibility of adopting LAPs for interpreting the model, especially as they only need a few epochs, even one, to train them.

| Dataset | Architecture | Org | WSL- | DLAP- | BotCL- | CRAFT |
|---------|--------------|-----|------|-------|--------|-------|
| RSNA | ResNet-18 | 41 s | 60 s | 42 s | 43 s | 30 min |
| | InceptionV3 | 102 s | 141 s | 70 s | 80 s | 42 min |
| ImageNet | ResNet-50 | 2 hr 15 min | 1 hr 45 min | 1 hr 23 min | — | 128 hr |
| | InceptionV3 | 2 hr 44 min | 1 hr 44 min | 1 hr 44 min | — | — |
| | EfficientNetB0 | 1 hr 25 min | 1 hr 16 min | 1 hr 16 min | — | 258 hr |

and different XAI methods in a Colab Notebook with a T4 accelerator. The results for 761 positive test cases are presented in Sec. D.4.2. It can be observed that LAP has been significantly faster than all the methods in all architectures, but one with a slight difference: GC on the original ResNet. As the complexity of the model increases, e.g. in Lap-extended Inception architecture vs the original ResNet, the execution time of the other XAI methods increases vastly, while LAP is not affected by that. To be more strict, if we compare the interpretation time of any other XAI method on the original architectures with the interpretation time of LAP in the LAP-extended architectures, we can conclude that if interpretation is consumed as an inseparable part in running the model, LAP-extension has led to saving time rather than proposing time problems.

Table 16: The pure "prediction + interpretation" time in seconds for 761 positive test cases in a Colab Notebook with a T4 accelerator for different XAI methods and different architectures. Run times are calculated for the vanilla ResNet18 (RN) and Inception3 (I3) models, BotCL-extended models (BCL-), LAP-extended models trained using the proposed weak supervision (WSL-), and LAP-extended models trained using expert annotations (BBL-). Comparing the time required for interpretation with LAPs in the LAP-extended architectures with the other methods in the original architectures indicates that LAP-extension has not introduced extra time cost considering interpretation as an essential aspect for the modern utilization of deep networks. All values are reported as mean $\pm$ standard deviation over five independently trained models initialized with different random seeds.

| Entry | DL | GGC | GC | GBP | RC | LAP | BotCL | CRAFT |
|-------|-----|------|-----|------|-----|-----|-------|-------|
| ResNet18 | $4.98 \pm 0.34$ | $4.14 \pm 0.18$ | $\mathbf{2.06} \pm 0.23$ | $3.21 \pm 0.33$ | $8.49 \pm 0.17$ | $2.52 \pm 0.23$ | NA [*] | $100.98 \pm 5.93$ |
| BCL-RN | $6.5 \pm 0.1$ | $5.83 \pm 0.1$ | $3.48 \pm 0.12$ | $4.71 \pm 0.15$ | $8.60 \pm 0.09$ | $3.05 \pm 0.2$ | $\mathbf{2.99} \pm 0.17$ | $138.02 \pm 5.47$ |
| WSL-RN | $8.19 \pm 0.11$ | $7.31 \pm 0.26$ | $4.03 \pm 0.31$ | $5.94 \pm 0.36$ | $10.32 \pm 0.14$ | $\mathbf{2.79} \pm 0.28$ | NA | $130.86 \pm 8.02$ |
| BBL-RN | $7.55 \pm 0.08$ | $6.71 \pm 0.23$ | $3.51 \pm 0.29$ | $5.33 \pm 0.47$ | $10.22 \pm 0.13$ | $\mathbf{2.8} \pm 0.32$ | NA | $130.02 \pm 3.12$ |
| Inception V3 | $23.58 \pm 0.59$ | $90.47 \pm 0.34$ | $23.06 \pm 0.14$ | $11.63 \pm 0.31$ | $35.57 \pm 0.37$ | $\mathbf{5.14} \pm 0.5$ | NA | RL [†] |
| BCL-I3 | $23.63 \pm 0.36$ | $87.53 \pm 0.19$ | $25.03 \pm 0.19$ | $13.71 \pm 0.21$ | $36.04 \pm 0.49$ | $8.15 \pm 0.44$ | $\mathbf{7.95} \pm 0.18$ | RL |
| WSL-I3 | $31.71 \pm 0.17$ | $146.85 \pm 0.28$ | $49.99 \pm 0.06$ | $20.04 \pm 0.19$ | $46.95 \pm 0.07$ | $\mathbf{4.86} \pm 0.21$ | NA | RL |
| BBL-I3 | $24.85 \pm 0.25$ | $119.31 \pm 0.15$ | $38.8 \pm 0.08$ | $16.28 \pm 0.19$ | $44.65 \pm 0.04$ | $\mathbf{4.72} \pm 0.21$ | NA | RL |
| Avg RN-* | $6.81 \pm 1.21$ | $6 \pm 1.19$ | $3.27 \pm 0.73$ | $4.8 \pm 1.01$ | $9.41 \pm 0.86$ | $\mathbf{2.79} \pm 0.19$ | $2.99 \pm 0$ | $124.97 \pm 14.2$ |
| Avg I3-* | $25.94 \pm 3.37$ | $111.04 \pm 24.12$ | $34.22 \pm 10.94$ | $15.42 \pm 3.14$ | $40.8 \pm 5.07$ | $\mathbf{5.72} \pm 1.41$ | $7.95 \pm 0$ | RL |

[*] Not Applicable: The BotCL interpretation method applies only to BotCL-extended architectures.
[†] The experiment was interrupted due to exceeding the resource limits in Google Colab.

The required inference time was also measured for different models on the ImageNet dataset. The results presented in Tab. 17 show that LAP is among the fastest methods. Having an efficient inference time and superior interpretative performance across various domains makes it a compelling choice for practical applications.

Table 17: The time required to run the model and interpret outputs on the ImageNet validation data was measured for various XAI approaches for interpreting the last convolutional layer as the target. LAP proved to be one of the fastest methods. Reasonable time combined with its superior performance, makes LAP a highly suitable choice for practical applications.

| Model | LAP | DL | GGC | GBP | RC | GC | CRAFT |
|---|---|---|---|---|---|---|---|
| Org ResNet50 | 179 | 276 | 188 | 137 | 523 | **127** | 500 |
| LAP-ResNet50 | 220 | 362 | 283 | 205 | 561 | **153** | 500 |
| Org InceptionV3 | 232 | 423 | 282 | 215 | 784 | **170** | -[*] |
| LAP-InceptionV3 | 230 | 438 | 296 | 224 | 788 | **171** | - |
| Org EfficientNetB0 | 127 | 138 | 135 | 108 | 186 | **105** | - |
| LAP-EfficientNetB0 | 120 | 143 | 135 | 125 | 186 | **104** | 230 |
| Average | 184.67±46.7 | 296.67±122.08 | 219.83±69.6 | 169±46.76 | 504.67±246.61 | **138.33**±28 | 410±127.28 |

[*] CRAFT was assessed on only three models, as training it requires substantial time (see Tab. 15).

### D.5  LAPs as standalone predictors

#### D.5.1  RSNA pneumonia detection

Each LAP layer can also be used as a standalone predictor. In pneumonia classification, if an LAP has assigned a probability of more than 0.5 to at least one pixel, it has found pneumonia in the image. The prediction of the LAP for these samples is assumed to be positive and otherwise negative. We evaluated the predictivity of LAP modules as standalone deciders and the compatibility of their predictions with the model's decisions. According to Tab. 18, as expected, the deeper the layer and the larger its receptive field, the higher the predictivity and the compatibility. Assessing LAPs allows us to diagnose the model from various directions. In this example, for an architecture as complicated as Inception V3, $LAP_3$ has nearly reached the performance of the whole model. Even in the BB versions, it surpasses the performance of $LAP_4$ and the final decider. We can decide that the layers after $LAP_3$ have suffered more from overfitting to the training data, resulting in a performance drop in the test data. Additionally, it can be observed that the BB-LAPs in both architectures generally have higher accuracies than WS-LAPs, especially in $LAP_2$. This observation implies that using exact supervision leads to better learning in shallower layers of the network, and the DNNs try to recover the performance in deeper layers. Notably, the adopted weakly supervised training scheme has resulted in the enhancement of the predictivity of LAPs in shallower layers ($LAP_2$) compared to the vanilla models, for which the LAPs are trained as post-hoc explainers over a frozen fully trained model.

Table 18: Results on the RSNA dataset for vanilla and LAP-extended models trained by the proposed weakly supervised loss (WS) and expert annotations (BB). F1 metrics of LAPs as stand-alone predictors w.r.t. ground truth (predictivity) and model prediction (model compatibility) provide an approach for the diagnosis of the model in a global sense. All values are reported as mean ± standard deviation over five independently trained models initialized with different random seeds.

| Model | Predictivity | | | | Model compatibility | | | |
|---|---|---|---|---|---|---|---|---|
| | $LAP_1$ | $LAP_2$ | $LAP_3$ | $LAP_4$ | $LAP_1$ | $LAP_2$ | $LAP_3$ | $LAP_4$ |
| ResNet 18 | 67.78±0.0 | 68.24±0.3 | 81.52±0.34 | 93.3±0.82 | 66.87±0.71 | 67.36±0.68 | 81.18±0.92 | 95.65±1.05 |
| WSL-RN | 71.05±2.85 | 85.61±4.59 | 92.78±1.66 | 93.26±1.16 | 71.29±2.26 | 86.63±4.3 | 95.45±1.69 | 96.29±1.2 |
| BBL-RN | 71.03±2.99 | 89.13±2.66 | 94.66±0.28 | 95.04±0.21 | 70.26±3.63 | 89.25±3.44 | 97.04±0.76 | 98.98±0.4 |
| Inception V3 | 67.78±0.0 | 81.86±2.0 | 95.17±0.35 | 93.76±1.29 | 66.41±0.43 | 80.59±2.52 | 97.98±0.24 | 96.41±1.11 |
| WSL-I3 | 67.78±0.0 | 89.51±1.0 | 95.21±0.09 | 95.27±0.11 | 66.54±0.27 | 89.13±1.55 | 98.33±0.44 | 98.53±0.45 |
| BBL-I3 | 67.78±0.0 | 94.02±1.17 | 95.3±0.13 | 95.24±0.13 | 66.85±1.01 | 94.88±0.69 | 99.61±0.17 | 99.7±0.14 |

#### D.5.2  ImageNet

In contrast to RSNA, which has a simple DC, ImageNet has 1000 DCs with objects of high-variance sizes. Therefore, evaluating the LAP's predictivity and model compatibility is not straightforward. We defined the DC size features, $F_C \in \mathbb{R}^{1000}$, as the sum of pixels' importance scores in each DC map. Then, we trained a 2-layer network to classify the samples according to $F_C$. To evaluate the predictivity and model

compatibility of the LAP, we compared the predictions of the mentioned MLP network with ground-truth and LAP-Extended ResNet 50 predictions, respectively. The results are presented in Tab. 19 for the LAP at layer 4 which uses the output of the layer 3 in the model. Despite the depth of the layer and its distance from the final decider, complexity of the prediction task, the high similarity between many of the classes in the domain, and the highly summarized information extracted from the DCs' score maps, the LAP has achieved high predictivity and model compatibility.

Table 19: The top-1 and top-5 accuracies have been computed for the original model (ResNet 50), and tuned LAP-extended model. The tuned model performs better than the original model and is also accommodated with self-interpretability. The accuracy of the LAP predictor over the output of layer 3 w.r.t. ground truth (predictivity) and the model's prediction (model compatibility) show its effectiveness despite its distance from the final layer.

| Model | Model performance | | LAP predictions | | Compatibility with the model | |
|---|---|---|---|---|---|---|
| | Top-1 Acc. | Top-5 Acc. | Top-1 Acc. | Top-5 Acc. | Top-1 Acc. | Top-5 Acc. |
| ResNet 50 | 76.13 | 92.86 | - | - | - | - |
| LAP ResNet 50 | **76.40** | 92.93 | 71.98 | 86.4 | 71.98 | 90.83 |

## D.6   More examples of LAP interpretations

Due to the limited number of pages in the main paper, we have provided more images interpreted with LAP for RSNA and Imagenet in Figs. 8 and 9, respectively.

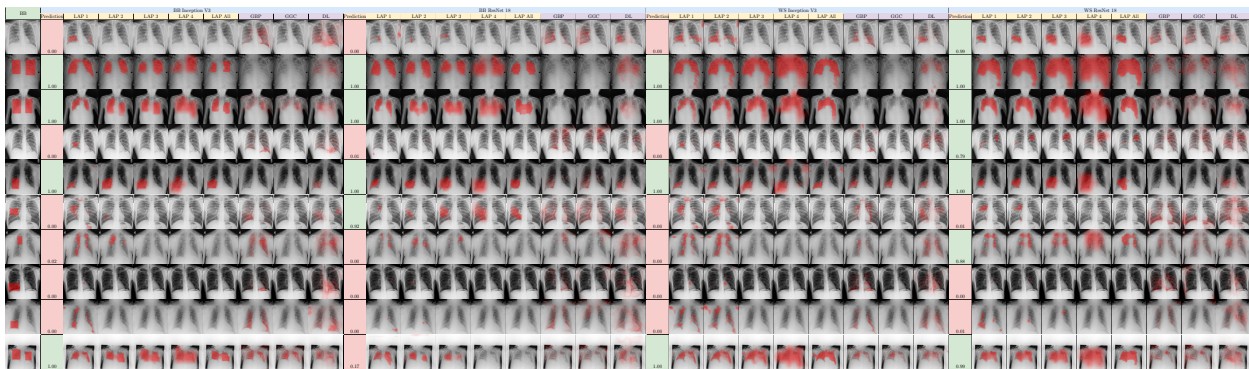

Figure 8: Examples of all models' interpretations (LAPs) compared to RSNA bounding boxes (BB) and other XAI methods (GBP, GC, GGC, DL, and RC) on the RSNA dataset. The probability assigned by each model is presented, and the background color represents the final decision of the model. Despite other methods, LAP interpretation is faithful to the model's prediction and helps diagnose the layer from which the model has made a mistake in decision-making.

## D.7   Further Related Literature

As the proposed LAP-Based Interpretation is a versatile approach—i.e., LAPs function both as plug-in modules for inherent interpretability and as detached modules for post-hoc explanation—the primary related literature for this work consists of white-box XAI methods that meet the following criteria: they interpret models with little or no modification; they do not require additional expert annotations, external data sources, or externally transformed concepts from other networks trained on datasets other than the primary task's own dataset; they are not specific to a particular family of tasks or models (e.g., transformers); and they do not incur high costs for training an extra model from scratch. Nonetheless, broader families of XAI methods can also be considered related to this work, and we summarize them in this section and discuss the differences in our approach.

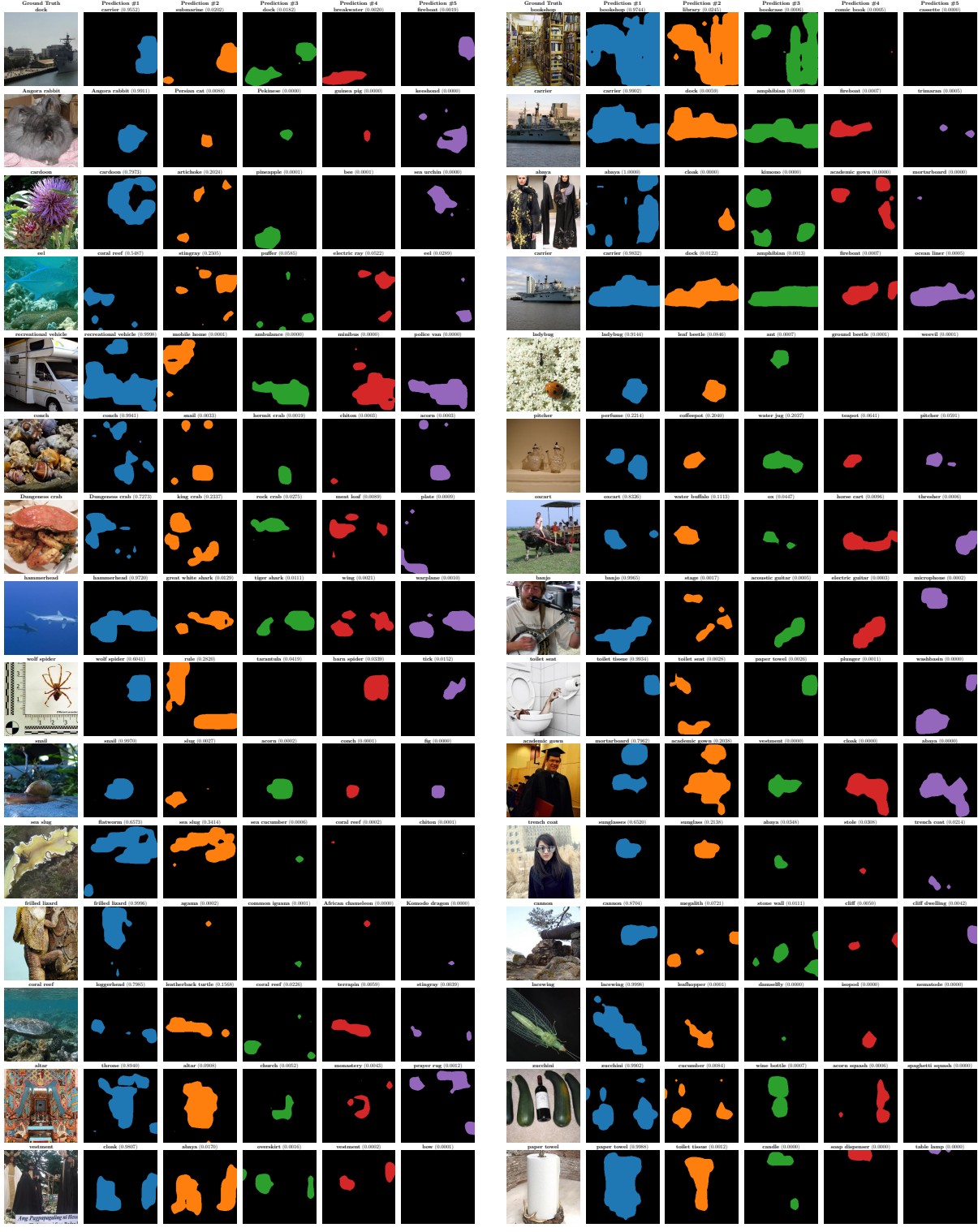

Figure 9: Further DC-wise interpretation Examples of LAP ResNet 50 on ImageNet dataset. For each example, DC heads related to the top-5 predictions of the model are illustrated separately. The probability assigned by the model for each DC head is presented, too. The attribution maps clarify why the model has chosen each class as one of its top 5.

**Surrogate Attribution Map Learning:** This class of post-hoc methods trains a separate proxy (surrogate) model to generate attribution maps for the primary network. In this line, (Dabkowski & Gal, 2017) framed the explanation as the minimal region that preserves the original prediction and trained a U-Net–style proxy with an objective aligned with this definition. (Chen et al., 2018; Yoon et al., 2018) adopted an information-theoretic perspective, casting explanations in terms of mutual information, information gain, and KL-divergence, and introduced a corresponding objective for training the proxy. (Schwab & Karlen, 2019) followed a similar approach, grounding the objective in identifying causal relationships between features and the output. (Schulz et al., 2020) performed per-instance optimization to find an information bottleneck over an intermediate feature map, reducing information flow while preserving the output, and subsequently converted the bottleneck into an explanation. (Jethani et al., 2021) proposed an objective for training a proxy function to estimate SHAPELY values (Lundberg & Lee, 2017), a theoretically grounded attribution method whose exact computation is exponentially expensive. (Covert et al., 2023) further expanded this line of work to make it efficient enough for application to ViTs. Subsequently, (Wang et al., 2025) proposed leveraging intermediate feature maps of the original network to train a side proxy network, rather than training a separate ViT from scratch, thereby further reducing both training and inference time.

The introduced methods all train a separate proxy model to approximate either an expensive explanation method or an optimization-based explanation. They typically rely on approximations in the objective, extensive sampling, and various forms of masked input data. Consequently, they suffer from training difficulties and imperfect approximations. In contrast, our method uses only the same training data as the main model, requiring no specific extra augmented data (e.g., masked versions), so it can be trained from scratch alongside the main model. The LAP modules are lightweight components that work with intermediate feature maps, making them far lighter than separate networks used as proxies. Additionally, our proposed framework, and its LAP-supervision objective, is directly fidelity-oriented, aligning it with the primary task rather than with estimating another explanation approach.

**Attention-Based Interpretation:** The attention mechanism offers an intuitive way to interpret networks by computing importance weights for different components. Certain domains or tasks, such as visual question answering (Anderson et al., 2018), image captioning (Anderson et al., 2018), and whole-slide image analysis in histopathology (Kapse et al., 2025), require attention as part of their solution design. In these cases, using attention weights as interpretations is valid when attention serves as the final decision-making layer, directly computing the embedding for the output. However, the underlying feature extractor beneath that layer remains a black box. In other architectures, such as transformers (Vaswani et al., 2017; Dosovitskiy et al., 2021), attention modules are distributed throughout the model. Studies have adopted attention weights to assess which token receives the most focus depending on the input (Clark et al., 2019). The challenge is that attention weights are distributed across multiple heads and layers, making it unintuitive which ones contributed to the final decision (Jain & Wallace, 2019; Serrano & Smith, 2019; Covert et al., 2023). To address this ambiguity, a line of work including "attention rollout" and "attention flow" has emerged to trace attention modules through the network (Abnar & Zuidema, 2020).

This family of XAI methods is only applicable to architectures that use attention, which is not true for all architectures or all problem domains. Moreover, even architectures that adopt attention weights at the final layer still do not have means to provide interpretation of the entire network. In contrast, LAP-based interpretation is a general approach that does not require specific network modules; it is applied to intermediate feature maps, enabling through-the-depth interpretation. As another distinction, standard attention produces a probability distribution over elements, meaning that the weight assigned to each element has no absolute meaning and cannot be compared across different weight maps. This becomes problematic when the size of relevant regions varies across samples, often causing interpretations to either include irrelevant areas or omit portions of the truly important regions. In contrast, LAP computes an absolute probability of importance for each element for each network output, eliminating this issue.

**Weakly Supervised Attribution Learning:** Methods in this family were primarily developed for tasks typically addressed by weakly supervised solutions, such as Multi-Instance Learning (MIL). Several studies adopted an attention mechanism to aggregate feature embeddings extracted from different patches in Whole-Slide Imaging (WSI) into a single feature map for the entire slide, and used the learned attention weights as interpretations (Ilse et al., 2018; Kapse et al., 2025; Ma et al., 2025). Similarly, (Wienholt et al., 2026)

followed a MIL approach to analyze high-resolution chest X-ray images by dividing them into non-overlapping patches and aggregating patch-level logits via averaging into a single final logit, thereby leveraging the inherent interpretability that this solution provided. In the context of weakly supervised anomaly detection, (Wargnier-Dauchelle et al., 2023) proposed a penalization loss for gradient-based attributions in negative cases to reduce unintended attributions, which in turn enhanced the training of the primary task. Another family of studies adopted transferred knowledge or concepts from external sources; these are discussed under the heading "Learning Attributions Using External Sources."

In addition to the fact that methods in this family only provide aggregation weights as the final tier of decision making, leaving the feature extraction backbone beneath still opaque, the same distinctions discussed for Attention-Based Interpretations also apply here. Furthermore, none of the enumerated methods propose a weakly supervised loss function similar to our approach for learning the attributions. They typically learn attention weights through the primary task, and the final classifier is trained on the aggregated feature vector. In contrast, our proposed approach performs no such aggregation of elements into a single feature vector. Our loss term is designed specifically for our introduced approach, in which the relatedness of each input element to each network output is modeled as a binary random variable to produce an importance map over the elements. Consequently, the loss is applied over single selected elements rather than an aggregated version.

**Learning Attributions Using External Sources:** Some studies have proposed methods for incorporating human annotations into training attributions. (Ross et al., 2017) introduced additional loss terms to guide gradients as explanations, thereby teaching the model to explain its predictions for the right reasons. Concept Bottleneck Models (CBMs) (Koh et al., 2020) are a family of self-interpretable models that employ a concept bottleneck layer after the feature extraction backbone and train the classifier solely based on concepts extracted from the input. The initial version of this family relied on expert annotations for all concepts required by the task (Koh et al., 2020), but later variants, such as that of (Yuksekgonul et al., 2023), developed methods to use transformed concepts from large models trained on large-scale datasets, such as CLIP (Oikarinen & Weng, 2022). Similarly, (Kapse et al., 2025) used GONCH (Lu et al., 2024) in their training, which encodes visual-language concepts for histopathology. (Belém et al., 2021) inferred noisy pseudo-labels for the existence of related concepts based on human-developed knowledge encoded as existing rules in an algorithmic decision-making task. (Hajialigol et al., 2023) trained a network for the multi-task objectives of document classification and attribution map calculation for the document, where pseudo-labels for each document and the desired attributions were derived by passing masked document content to FLAN-T5 (Chung et al., 2024) and regenerating masks using attributions derived from BERT (Devlin et al., 2019) over multiple iterations.

Unlike our proposed framework, this family of methods depends on external sources for training. Expert annotations are costly and difficult to gather for large datasets. While transformed annotations from external models may make this approach feasible for general-purpose applications, they may not encode the specific concepts required for specialized tasks, such as those in rare medical domains.

**Attribution by Auxiliary Heads:** Most methods in this category designed a model for multi-task learning, with one branch performing the primary task and another branch generating attribution maps. (Peng et al., 2024) proposed adding an extra attribution branch to ProtoPNet (Chen et al., 2019), to be jointly trained with the primary branch. The auxiliary head shared the backbone encoder with the primary task's head and was trained using various masked inputs and the degree to which the masked inputs preserved the original output. (Preechakul et al., 2020; Phuangthongkham et al., 2023) also added an attribution generation branch to the main task's architecture to generate high-resolution Class Activation Maps at inference time. (Hajialigol et al., 2023) trained a network for the multi-task objectives of document classification and attribution map calculation for the document, where pseudo-labels for each document and the desired attributions were derived by passing masked document content to FLAN-T5 (Chung et al., 2024) and regenerating masks using attributions derived from BERT (Devlin et al., 2019) over multiple iterations. While not exactly the same category, (Wang et al., 2025) adopted the intermediate feature maps of the main task to train a lighter side network for generating SHAPELY values.

Each of the methods in this family employs a different solution for training the auxiliary head for attribution generation, whether using masked inputs to assess output consistency, annotations derived from external sources, or other XAI methods. Among the reviewed literature, our proposed method is the only one that

directly adopts a weak-supervision-style training approach to learn attribution maps based solely on its own predictions, without relying on any external sources.

**Weakly Supervised Semantic Segmentation:** Several studies have adopted XAI methods to provide annotations for the task of semantic object segmentation based solely on class labels, without relying on detailed annotations (Wang et al., 2022; Lee et al., 2021b; Somani et al., 2025; Aasem & Javed Iqbal, 2024; Kumar & Murugan, 2025). We argue, however, that the task of providing interpretations or explanations for a model is fundamentally different from object segmentation. As illustrated in Fig. 1, in XAI, the goal is to identify the input regions that a model has relied upon for decision-making, which may constitute only a part of the object (e.g., a rabbit's tail and ears). In contrast, object segmentation aims to detect the entire object and therefore follows a different objective.

## D.8 Importance based pooling

Pooling layers and strided convolutions are widely used in CNNs to increase the receptive field and decrease memory consumption. Gao et al. have proposed a unified framework for formulating different pooling strategies, called Local Aggregation and Normalization (LAN), and a pooling method called Local Importance Pooling (LIP) (Gao et al., 2019). This framework aggregates features within local sliding windows by weighted averaging. The weights are assigned based on the importance of the features. According to this framework, average pooling assumes the same importance score for all the pixels and is susceptible to feature fading. Max pooling assigns one to the highest feature and zeroes to all the others, leading to sparse gradient paths and slow training. Strided convolutions give importance based on the pixel's location in the window and are more sensitive to shift variances. LIP has used the attention mechanism to assign importance weights to the features. LIP is applied feature-wise, which makes it different from our proposed architecture. All of the mentioned pooling layers except strided convolutions can lead to loss of relative spatial relations, as they select different features from different spatial locations (Patrick et al., 2022).

## D.9 Licenses of the assets

- **RSNA**: The dataset is freely available for download from Kaggle. Based on the competition website "the competition data is allowed to be used for the competition, participation on Kaggle website forums, academic research and education, and other commercial or non-commercial purposes as long as the attribution for the dataset and the individual items (sound files) are provided when required."

- **ImageNet**: The dataset of ImageNet Large Scale Visual Recognition Challenge 2012-2017 is freely available for download from the ImageNet website. Based on the website, "researchers shall use the Database only for non-commercial research and educational purposes."

