# OpenReview forum: "Enhancing Interpretability: A Versatile Clue-Based Framework for Faithful In-Depth Interpretations"
_TMLR — Rejected by TMLR_

### Review · Reviewer_MjLx · 2026-03-16

**Summary Of Contributions:**

This paper addresses the challenge of, given a data set of images and a convolutional neural network, devising a method to identify image regions that are relevant to how the neural network classifies those images. I usually refer to this problem as the input feature attribution problem for image classifiers. This paper frames the problem as one about identifying distinguish includes (DCs) and introduces a local attention perception layer which can be trained to identify DCs. They also introduce a method for training neural networks with LAP layers to make them more intrinsically interpretable. As the main substantive contribution, this paper demonstrates that their method can beat base lines from the Captum library for identifying regions the human annotator's marked for image net and a medical data set.

**Additional Comments:**

I think that in a handful of ways, the writing is improvable. For example, there are some things that are pretty unclear. There's a grammatical error in one of the abstract sentences. The second key contribution is really not a clear or precise sentence. If I were advising this paper I would recommend doing a lot of work to be more concise and focused and clear. I think this fairly throughout the text.

I think that the paper also takes an unconventional approach to how it uses, and sometimes abuses, terms that have widely accepted definitions that are not in line with what is going on here. For example, I think that this paper uses its own implicit definitions of "interpertability" and "explainability" which are meaningfully different than the common ones. Meanwhile, one sentence I noticed "LAP can be viewed as an attention mechanism, since it assigns importance scores to input pixels" which is very confusing.

Overall, I found the paper confusing and hard to understand. If the other reviewers found it easy to understand, I will eat my words. But for what it's worth, I find it difficult to describe what I think is confusing about descriptions of the methods because I am not able to figure out what questions to ask. Overall, I think that the use of figures, descriptions, jargon, and notation have been ineffective. I also wish that the paper had an algorithm somewhere. For context, I have worked on image attribution and explainability methods in the past. I'm not up-to-date on the literature, but I'm familiar with the concepts. I'm still not able to understand this paper thoroughly from a single read through.

**Audience:**

No

**Audience Explanation:**

Table 2 seems like the strongest part of the paper, but I'm used to seeing tables like table 2. Methods like this have been studied for a very long time, and I think that I stopped being impressed by results like this in 2022 or 2023. I know that is cynical of me, and I don't think this criticism of mine is making the difference in my overall review. But I can honestly say that I think this paper is the kind of paper that I would've expected to see five years ago. I'm just not sure how much it does to really engage with unsolved problems. Again, I'm about to sound very cynical, but if I am being honest, I would not usually recommend people work on papers like this in 2026.

I think that paper is in 2026 which work on these types of problems with these types of models and these types of data sets should be able to go substantially further than beating some fairly simple pre-implemented baselines on some very old benchmarks. It's difficult for me to say what kind of contribution I think would be helpful here, because I think that this field has been somewhat "over-researched" already. But I think that something more would be needed. Something to demonstrate more proximal practical value, rather than another paper introducing another pixel saliency method that beats the Captum baselines.

**Claims And Evidence:**

No

**Claims Explanation:**

It seems that figure one doesn' contain results from an experiment but just has manually selected regions. If the method in this pape works, I ask myself, "why not put its DCs in figure 1?"

Definition 1 seems underspecified to me. What is a region? What is the attribution method by which we determine "favoring"? And favoring does not seem it could  be a binary thing in reality.

Section 2.1 seems fishy to me. After reading it a couple times I cannot gather what the counter factual being considered implicitly here is. When we consider an image without one of its DC's, what is taking its place? I know that the method for training LAPs later implicitly fixes this interpretation, but if I understand, section 2.1 isn't currently intelligible without context that happens later on in the paper.

Unless I misunderstand something, none of the baselines tested involved attention-based methods. So it seems like an unfair comparison.

Agreement with human annotations about salient regions in an image is valid. But it's worth noting that this is not necessarily the gold standard. Especially for imageNet. Methods that do particularly well at this are likely overfitting to how humans perceive images. Not what features models are able to classify them by. For example, DNNs can do surprisingly well on ImageNet by only seeing border pixels.

**Requested Changes:**

None

---

### Review · Reviewer_C7iJ · 2026-03-28

**Summary Of Contributions:**

This paper proposes a framework for interpreting deep neural network decisions using a concept called Distinguishing Clues (DCs). A DC is defined as the set of input regions that favor a subset of network outputs over others basically regions in the input that distinguish one class from another. The paper proposes that each class has a set of DCs and introduces a Background Only (BGO) class for regions that do not relate to any specific class. The completeness condition ensures every input is mapped to a DC but not necessarily to a particular class.

The paper then introduces the Local Attention Perception (LAP) module which sits on top of a chosen layer and produces importance scores for each spatial location in that layers feature map. It takes the feature vector at each spatial location and applies a simple 1x1 convolution followed by a sigmoid to produce a probability score between 0 and 1 for each DC at each location. The paper also introduces LAP-Pool which replaces average pooling with importance weighted pooling where important pixels contribute more. Multiple LAPs are applied at different layers and aggregated from deep to shallow to get a final map that is both accurate and spatially precise.

The paper introduces 2 options for using LAP:
- **Post-hoc**: frozen network produces feature maps and LAP reads them without touching the original model
- **Intrinsic**: LAP replaces pooling layers and is trained jointly from scratch

Since no pixel wise ground truth is available the paper proposes a weakly supervised training scheme. For any image the DC related to the ground truth class is present and all others are absent. This gives rise to 3 loss terms:
- **MinAR**: pushes top K pixels of the present DC toward 1 ensuring coverage
- **MaxAR**: pushes bottom K pixels of the present DC toward 0 acting as a sparsity constraint
- **IAR**: pushes top K pixels of absent DCs toward 0 focusing on the most confidently wrong pixels rather than wasting gradient on already correct ones  top K is used instead of all pixels to maintain balance with MinAR

## Experiments

**Baselines**: GBP, GradCAM, GuidedGradCAM, DeepLIFT, RelevanceCAM, CRAFT, BotCL

**Datasets**: RSNA Pneumonia Detection and ImageNet

**LAP Variants**: Weakly Supervised (WSL) using only class labels and Bounding Box supervised (BBL) using expert annotations

For the RSNA dataset they use IoU between binarized interpretation maps and expert bounding boxes on correctly predicted positive cases. LAP outperforms all baselines in IoU for both post-hoc and intrinsic settings with WSL performing close to BBL despite using no pixel annotations. They also show qualitative examples of how LAP changes across layers and can diagnose where the model made the wrong decision. Ablation studies motivate the choice of loss terms.

For ImageNet they start with pretrained models across ResNet50, InceptionV3 and EfficientNet and evaluate faithfulness by keeping k% of highest scored pixels zeroing the rest and re-evaluating the model. LAP outperforms baselines in 10 out of 12 cases and was marginally outperformed by GradCAM in the remaining 2.

## Strengths

- **Originality**: Good - the idea of defining DCs in an unsupervised manner and using an attention module to identify importance is novel. The weakly supervised training scheme is also novel and the flexibility to work both post-hoc and from scratch is great
- **Presentation**: Good - the paper is well written and easy to follow with the method clearly described and well motivated
- **Significance**: Good - the paper addresses an important problem in interpretability and provides an interesting and practical solution that requires no expert annotations


## Weaknesses

- **Soundness**: Fair
  - More recent inherently interpretable models like CBMs were not compared against which is a relevant and important baseline
  - For the ImageNet experiment the removal of pixels might have an adversarial effect on the network leading to potentially misleading faithfulness results [1]
  - The claim that LAP improves accuracy is hard to support since all models seem to be on par with each other and the gains are minimal though notably no accuracy drop is still a positive result
  - The claim about strong accuracy when retaining only top 10-30% of pixels is questionable since absolute accuracy at that retention level is around 20% which is quite low


[1] A Benchmark for Interpretability Methods in Deep Neural Networks

**Audience:**

Yes

**Audience Explanation:**

The paper addresses an important problem which is explaining black box neural networks which is very important in critical applications. The solution provided is pretty flexible it can be trained from scratch or post hoc giving similar downstream performance while providing good explanation. Additionally the fact that it works without expert annotations (WSL) makes it applicable to a much wider range of real world datasets where pixel level annotations are expensive or infeasible.

**Broader Impact Concerns:**

The paper does not include a broader impact statement. While the work is broadly positive in its goal of making neural networks more interpretable, there is a risk of misplaced trust where clinicians or practitioners over-rely on LAP highlighted regions without questioning the underlying model decisions especially in safety critical applications like medical diagnosis

**Claims And Evidence:**

Yes

**Claims Explanation:**

The paper evaluates across 2 datasets and multiple architectures (ResNet, InceptionV3, EfficientNet) with multiple baselines in terms of accuracy the method is on par with others and in terms of interpretability it seems to be doing better. More specifically:

- **Statistical rigor**: Results are reported as mean ± std over 5 independent runs with different random seeds and paired t-tests (p-value ≤ 0.001) are used to show significance of interpretability results which strengthens the claims considerably
- **Multiple evaluation metrics**: For RSNA they use three different binarization approaches (µ+σ, TopK selection, learned threshold) and LAP wins across all three making the interpretability claim more convincing
- **Ablation studies**: They motivate every component of the loss through ablations showing that removing any term degrades performance which supports the design choices

**Requested Changes:**

- The claim that LAP improves accuracy is overclaimed since all models seem to be on par with each other and the gains are minimal
- The claim about strong accuracy when retaining only top 10% of pixels is questionable since absolute accuracy at that retention level is around 20% which is not particularly convincing

---

### Review · Reviewer_hwUe · 2026-04-08

**Summary Of Contributions:**

This work designs a clue-based approach for neural network interpretability. It defines Distinguishing Clues (DC) as input regions that distinguish one output with others to interpret the model’s decision-making mechanism. It then proposes the Local Attention Perception (LAP) module to produce DC specific probability maps at chosen network depths. The method can work as a detached post hoc explainer or as a plugged-in component during training. One of the main contribution is the introduced weakly supervised training loss for LAP. Specifically, it uses coverage, precision, and exclusion constraints. These are implemented with MinAR, MaxAR, and IAR losses. This enables learning without pixel level expert annotations. The empirical study evaluates both a medical dataset with expert boxes and ImageNet. On RSNA, LAP maps align better with bounding boxes than several common explainers, across architectures and thresholding rules. On ImageNet, the paper reports higher prediction retention under masking, especially for intermediate layers. The work also argues that LAP style supervision can improve interpretability without degrading accuracy.

**Audience:**

Yes

**Audience Explanation:**

The submission works on a meaningful track in of machine learning (e.g., explainable AI), so some individuals might be interested in the topic this work covers. The idea of DC and the LAP provide a principled method to produce class-specific importance maps, and this approach is designed to work at different depths of a neural network instead of only at the input level. This depth-wise explanation can be meaningful for debugging representation learning and for analyzing where discriminative evidence emerges.

The weakly supervised training loss is also relevant and novel to the community because it targets to resolve a common practical bottleneck (e.g., the lack of pixel-level supervision for explanations). By using coverage, precision, and exclusion-based constraints instead of expert labels, this method reduces human annotation cost while still guiding attention maps toward discriminative regions. This is particularly attractive for domains like medical imaging or industrial detection, where localized annotations are expensive and where interpretability is often critically required.

Furthermore, the experimental results on a dataset with expert bounding boxes, such as RSNA pneumonia, make the work actionable for applied machine learning researchers. Quantitative localization comparisons against common explainers provide a clear signal that the learned maps can align better with expert regions, and ablations indicate the training objectives matter rather than being cosmetic. Readers interested in evaluation methodology for explanations will also care about how different binarization and thresholding choices affect reported IoU and related metrics.

Meanwhile interest would not be constrained to applications. The paper’s plugged-in versus post-hoc interpretability framework will attract discussion among AI community about what it means to build interpretability into a model versus explaining a fixed model after the prediction.

**Broader Impact Concerns:**

My major concern is over trust of the model predictions. More visually plausible saliency maps can encourage users, especially in high stakes areas such medical imaging and diagnosis, to treat the model as reliable even when the prediction is wrong. The work should explicitly emphasize that explanation quality does not imply correctness, and recommend human-level expert in the loop use, decision audits, and clear guidance on how explanations should and should not be used in clinical workflows.

**Claims And Evidence:**

Yes

**Claims Explanation:**

For the empirical study on RSNA pneumonia, the work provides convincing evidence because explanations can be compared against expert bounding boxes. The authors as across multiple backbone architectures and compare against several established attribution baselines, and they report results over multiple random seeds. They also include ablations showing that the proposed weak supervision objectives materially affect localization quality rather than producing similar maps by default. Ablations on the loss components and hyperparameters, with sensitivity analyses showing when naive settings degrade interpretability.

However, the evidence is less conclusive for the claim that the method is generally more faithful on large scale natural image recognition. The ImageNet evaluation relies primarily on a retention style protocol where only the top \(k\%\) highlighted pixels are kept and the rest are masked. This type of evaluation can be sensitive to the masking and can introduce distribution shift. The higher retention does not necessarily imply the highlighted region is the true causal evidence. In addition, some variants involve adding modules and fine tuning, which can confound interpretability gains with changes to the underlying predictor and its robustness.

Besides, on ImageNet, LAP benefits from additional supervised training while many baselines are post-hoc explainer with very few training. This unfairness should be acknowledged and mitigated by including trained self-explaining baselines (e.g., for better comparison fairness.

**Requested Changes:**

-	The post-hoc explainer still requires dataset-level training of additional modules, which differs from lightweight post-hoc explainers. The paper could be more explicit about this tradeoff. For example, the submission will benefit from including more detailed efficiency analysis or algorithm complexity.
-	Details of cross-depth aggregation (App. A.5), consistency losses, and the selective fine-tuning regime are central but relegated to appendices. A concise main-text summary would improve readability of these content.
-	The submission can strengthen faithfulness evaluation on ImageNet. The current retention test mainly uses one masking scheme, which can be sensitive to OOD artifacts and may not reflect causal relationships.
-	It is recommended to clarify the gap between the formal DC definition and the implemented training objective. The paper defines DCs formally, but the implemented supervision uses Top K and Bottom K heuristics and auxiliary constraints, so the mapping from theory to practice is not explicit.

---

### Review · Reviewer_NsSW · 2026-04-20

**Summary Of Contributions:**

The paper proposes a local interpretation framework based on Distinguishing Clues (DCs) and a Local Attention Perception (LAP) module. The method can be used in two modes: as a detached post-hoc explainer for frozen pretrained networks, or as a plug-in module trained jointly with the backbone. The paper also introduces a weakly supervised training objective based on MinAR / MaxAR / IAR losses to learn DC-wise importance maps without dense expert annotations. Empirically, the method is evaluated on RSNA pneumonia detection and ImageNet, and the authors report improved localization / faithfulness metrics relative to several saliency and concept-based baselines.

Key strengths:

- The paper is ambitious in scope: it aims to unify intrinsic and post-hoc explanation within one framework.
- The empirical section is fairly extensive, including multiple architectures, two datasets, ablations, and comparison to several baselines.
- The weakly supervised objective is reasonably well motivated and appears more thoughtful than a naive top-k supervision scheme. The ablation study suggests that the exact loss design matters.

Key weaknesses:

- In my view, the main methodological idea is not highly novel. Adding an attention-like module that predicts importance scores, together with an auxiliary or weakly supervised loss, is already a fairly established pattern in interpretable/self-explaining neural networks.
- The paper makes strong claims about faithfulness and interpretability, but the post-hoc setting is not fully convincing from that perspective. In the detached setting, the backbone is frozen and only the LAP head is trained afterward; therefore, the learned importance map is not itself part of the mechanism that produced the original prediction.
- As a result, the method may be identifying features that are predictive of the target class, or features that allow the detached head to preserve the model’s output under masking tests, rather than the exact internal evidence actually used by the original predictor. This weakens the paper’s strongest interpretability claims.

**Audience:**

Yes

**Audience Explanation:**

Yes. I think this paper would be of interest to readers working on interpretability, saliency methods, concept-based explanations, and self-explainable architectures.

The paper addresses a real and important question: how to obtain localized, potentially multi-layer explanations without dense annotations, and how to bridge post-hoc and intrinsic interpretability within one framework. The empirical comparisons on RSNA and ImageNet are also broad enough that some readers will likely find the results informative.

**Claims And Evidence:**

No

**Claims Explanation:**

The paper does provide nontrivial empirical evidence for its method. In particular, it shows improved IoU against expert bounding boxes on RSNA and stronger masking / SRG-style faithfulness results on ImageNet relative to several baselines. It also includes ablations suggesting that the proposed weak supervision is more effective than simpler alternatives. These are meaningful positives.

However, I do not think the evidence fully supports the paper’s stronger claims about **faithful interpretability**.

My main concern is conceptual. In the post-hoc setting, LAP is trained after the fact on a frozen network, and the paper explicitly states that detached LAPs explain pretrained models without modifying their architecture or weights. In that setting, the learned importance scores are not actually used by the original model when producing its prediction. Therefore, even if those scores correlate well with regions that preserve the prediction under masking, this still does not establish that LAP has recovered the true internal evidence or exact importance weighting used by the model itself.

Put differently, the paper seems to conflate:
1. finding regions that are highly predictive of the target or useful for reconstructing the prediction, and
2. identifying the actual features and weights the original model relied on internally.

Those are related, but they are not the same. This distinction is especially important because the paper repeatedly frames the method as more faithful than alternative explainers. The current experiments do not fully close that gap.

I also found the novelty claims somewhat overstated. The central template — adding an attention-like module that outputs importance maps, then shaping these maps with auxiliary losses or weak supervision — does not feel fundamentally new. The paper’s more specific contributions are the DC formalization and the particular training recipe, but these are, in my opinion, closer to an incremental refinement than a major conceptual advance.

So overall, I think the paper contains useful experiments and some technically interesting ideas, but the evidence is not yet fully convincing for the level of interpretability / faithfulness / novelty claimed in the submission.

**Requested Changes:**

1. **Tone down the interpretability / faithfulness claims, especially for the detached post-hoc setting**.

 The paper should clearly distinguish between:
 - identifying regions whose preservation maintains the prediction, and
 - recovering the actual features / weights used internally by the original model.
 In the detached setting, the LAP scores are learned after training and are not part of the original prediction mechanism, so calling them faithful model-internal explanations is too strong as currently written.

2. **Clarify the conceptual status of LAP relative to attention and explanation**.

The paper states that LAP is attention-like but distinct from standard attention. That distinction should be made sharper, and the authors should explicitly discuss why an attention-like score map learned post hoc should be considered interpretability rather than merely a learned attribution surrogate.

3. **Substantially revise the novelty positioning**.

The paper should more candidly place itself relative to prior work on attention-based self-interpretability, auxiliary explanation heads, and weakly supervised attribution learning. The contribution seems to be a particular formalization and training scheme, not the introduction of a fundamentally new paradigm.

4. **Add experiments or analyses that better address the “used by the model” question**.

For example, the paper could compare detached LAP against mechanisms that are actually on the causal path of prediction, or otherwise provide stronger evidence that LAP is recovering model-used evidence rather than merely post-hoc class-correlated evidence.

---

### Decision · Action_Editor_wmvg · 2026-05-11

**Recommendation:** Reject

**Audience:**

Yes

**Audience Explanation:**

Three of the four reviewers (NsSW, hwUe, and C7iJ in their original review) confirmed audience interest. Local feature attribution, weakly supervised attribution learning, and the boundary between intrinsic and post-hoc interpretability remain active discussion areas in the TMLR community. The clinical RSNA evaluation with expert bounding boxes also offers a relevant testbed for applied interpretability researchers. Reviewer MjLx's "No" vote on audience reflects an assessment of contribution significance rather than topical relevance.

**Claims And Evidence:**

No

**Claims Explanation:**

Two of the three reviewers who submitted final Official Recommendations indicated that the claims are not adequately supported (Reviewer NsSW: No; Reviewer MjLx: No). The substantive concerns that remain unresolved after the rebuttal are the following:

1. Conceptual gap in the faithfulness claim for the post-hoc setting. Reviewer NsSW notes that in the detached configuration, LAP is trained after the fact on a frozen network, so its importance scores are not part of the original prediction mechanism. The paper conflates "finding regions whose preservation maintains the prediction" with "identifying the actual features and weights the model relied on internally." Although the rebuttal added an SRG evaluation and softened the language throughout the manuscript, the reviewer's final position is unchanged: "the novelty is somehow lacking, and I'm not fully convinced about the story." The proposed rename from "Local Attention Perception" to "Local Attribution Perception" is a cosmetic change and does not address the underlying issue.

2. Underspecified formal definition and unclear counterfactual. Reviewer MjLx points out that Definition 1 does not clearly specify what constitutes a "region" or what "favoring" means, that the implicit counterfactual in Section 2.1 is unintelligible without context introduced later in the paper, and that the connection between the formal Distinguishing Clue definition and the implemented Top-K / Bottom-K training heuristic is not rigorous. The author response clarifies the intended reading but does not fully repair the underlying formalization in the manuscript.

3. Limited methodological novelty relative to prior work. The central template — adding an attention-like module that predicts importance scores, combined with auxiliary or weakly supervised losses — is a well-established pattern in self-explaining networks. Reviewer MjLx's firmly negative assessment notes that the work "is the kind of paper I would've expected to see five years ago" and "does not engage with unsolved problems." The rebuttal added a new "Further Related Literature" appendix but did not produce a sharper methodological differentiation from existing weakly supervised attribution and attention-based interpretability approaches.

4. Evaluation framework remains contested. While the rebuttal adds SRG scores (LAP averages 0.48 vs 0.42 for the next-best Relevance-CAM on ImageNet), the gain is modest and the comparison still leans on ImageNet pixel-flipping protocols that multiple reviewers flagged as susceptible to OOD artifacts. Reviewer hwUe's concern about training-budget fairness across baselines was only partially resolved in follow-up.

The cumulative weight of these concerns — particularly the firm Reject from Reviewer MjLx that was reaffirmed twice after seeing the revisions, and Reviewer NsSW's unresolved Leaning Reject — indicates that the evidence does not adequately support the paper's stronger claims about faithfulness and methodological advance.